# CODECHAIN: TOWARDS MODULAR CODE GENERATION THROUGH CHAIN OF SELF-REVISIONS WITH REPRESENTATIVE SUB-MODULES

**Hung Le, Hailin Chen, Amrita Saha, Akash Gokul, Doyen Sahoo, Shafiq Joty**
Salesforce Research
{hungle, hailin.chen, amrita.saha}@salesforce.com

## ABSTRACT

Large Language Models (LLMs) have already become quite proficient at solving simpler programming tasks like those in HumanEval or MBPP benchmarks. However, solving more complex and competitive programming tasks is still quite challenging for these models - possibly due to their tendency to generate solutions as monolithic code blocks instead of decomposing them into logical sub-tasks and sub-modules. On the other hand, experienced programmers instinctively write modularized code with abstraction for solving complex tasks, often reusing previously developed modules. To address this gap, we propose CodeChain, a novel framework for inference that elicits modularized code generation through a chain of self-revisions, each being guided by some representative sub-modules generated in previous iterations. Concretely, CodeChain first instructs the LLM to generate modularized codes through chain-of-thought prompting. Then it applies a chain of self-revisions by iterating the two steps: 1) extracting and clustering the generated sub-modules and selecting the cluster representatives as the more generic and re-usable implementations, and 2) augmenting the original chain-of-thought prompt with these selected module-implementations and instructing the LLM to re-generate new modularized solutions. We find that by naturally encouraging the LLM to reuse the previously developed and verified sub-modules, CodeChain can significantly boost both modularity as well as correctness of the generated solutions, achieving relative pass@1 improvements of 35% on APPS and 76% on CodeContests. It is shown to be effective on both OpenAI LLMs as well as open-sourced LLMs like WizardCoder. We also conduct comprehensive ablation studies with different methods of prompting, number of clusters, model sizes, program qualities, etc., to provide useful insights that underpin CodeChain's success [1].

## 1 INTRODUCTION

It has been a long-standing goal in AI to develop systems that can generate executable and functionally correct computer programs to solve complex problems (Manna & Waldinger, 1971). In recent years, we have witnessed unprecedented progress in this field, specifically with the remarkable success of large pretrained language models or LLMs (Koubaa, 2023; Wang & Komatsuzaki, 2021; Radford et al., 2019). Originally developed for natural languages, these models have been extended with a combination of code and text modeling capabilities (Rozière et al., 2023; Black et al., 2021; Chen et al., 2021), resulting in good performance in code generation from natural language problem description (Li et al., 2023; Luo et al., 2023; Wang et al., 2023). However, when evaluated on highly complex coding tasks, the current SoTA models still cannot match a skillful developer (Hendrycks et al., 2021; Li et al., 2022; Shinn et al., 2023), mostly due to their naive generation approach.

Most prior approaches with LLMs adopt a naive generation method in which the models would typically generate the code solution as a single monolithic block of code instead of decomposing the task into logical sub-tasks. Another limit of this naive generation approach is that the models would simply generate a large number of solutions independently, with the hope that one of the solutions

---

[1]https://github.com/SalesforceAIResearch/CodeChain

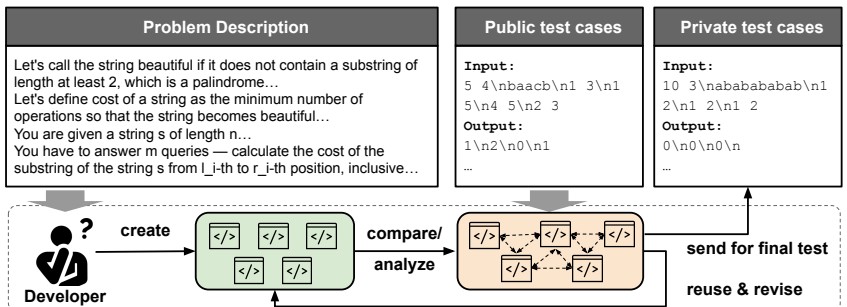

Figure 1: *[Top]* An example of a code generation task from CodeContests (Li et al., 2022) where the problem description and public test cases are provided as inputs to the model. *[Down]* We illustrate a typical problem-solving process in which a developer attempts to solve the problem iteratively, revising and reusing parts of their previously developed codes until satisfied.

would pass all the private test cases (Chen et al., 2021; Li et al., 2023; Austin et al., 2021). More recently, Li et al. (2022); Chen et al. (2023b); Zhang et al. (2023b) propose to sub-sample output programs using some forms of feedback from the public test results. However, these approaches assume that the sub-sampled programs could pass the private test cases, even without revising or debugging the programs. Some recent works like (Zhang et al., 2023a; Olausson et al., 2023; Le et al., 2022; Chen et al., 2023c;a; Shinn et al., 2023) have addressed this by performing self-revision with LLMs, utilizing feedbacks such as compiler error messages, test outcomes, and natural language explanation to improve the generated solutions. However, these approaches limit to using only independent feedback from individual solutions, neglecting potential collective insights from all generation samples or their sub-components.

On the other hand, in today's agile development environment, experienced developers are fully familiar with the concept of modularity in programming. Given a problem, they would instinctively write solutions that are modularized by high-level logical sub-tasks and sub-modules. The developers would then keep testing and analyzing their implementations, altering modular components from their previously developed solutions to efficiently improve their final solutions (see Figure 1). Inspired by this problem-solving process, we propose **CodeChain**, a novel inference framework to improve code generation in LLMs through a chain of sub-module based self-revisions (see Figure 2).

Specifically, in CodeChain, to incorporate modularity in code generation, we first introduce chain-of-thought prompting to instruct LLMs to decompose their solutions into modular segments. Each modular segment represents an abstract function that is intended for a high-level logical sub-task. To leverage this modularity in programs, we propose to further improve the generation process through a chain of self-revisions, each of which is conditioned by a set of sampled sub-modules as follows: (i) we first extract the sub-modules found in generated programs and group them into clusters. Within each cluster, we sample the centroid sub-modules and treat them as representative and reusable code parts for self-revision. (ii) We then augment the original chain-of-thought prompt with these selected sub-modules and instruct LLMs to generate new modularized solutions. With this approach, LLMs can receive the collective insights from modular components of all past generation samples to improve their future generations, imitating the problem-solving process of an experienced developer.

Our experiments show that CodeChain can significantly boost LLM performance and achieve SoTA performance on challenging code tasks in APPS (Hendrycks et al., 2021) and CodeContests (Li et al., 2022). Concretely, CodeChain improves the average *pass@1* performance by more than 35% on APPS and 76% on CodeContests. We also observe consistent improvements for both OpenAI LLMs as well as open-sourced LLMs such as WizardCoder (Luo et al., 2023). We further conducted comprehensive ablation studies, including analysis in single vs. multi-step revisions, feedback types, number of clusters, etc., and derived useful insights behind CodeChain's success.

## 2 RELATED WORK

Broadly related to our work is the research of large Transformer-based language models (LLMs) (Koubaa, 2023; Brown et al., 2020; Radford et al., 2019; Wang & Komatsuzaki, 2021; Touvron et al., 2023a). Originally designed for natural language processing, these models have been extended to

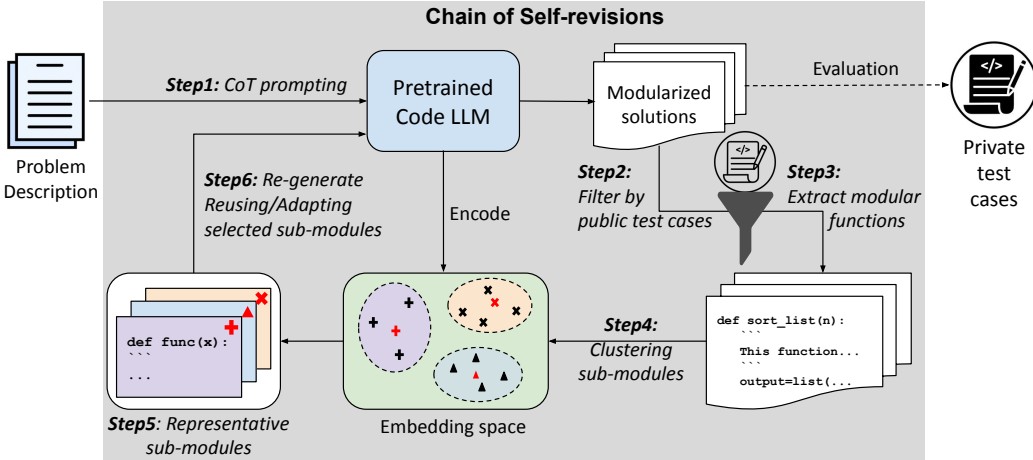

Figure 2: An overview of CodeChain: a pretrained LLM is first instructed with chain-of-thought prompting to generate a set of modularized solutions. Generated sub-modules are then extracted from potentially correct solutions and grouped into different semantic clusters. The cluster centroids are selected as representative sub-modules to condition the next self-revision round. The model is instructed to reuse or adapt these modules into its revised solutions.

learn from large-scale code data and become proficient in understanding contexts and generating outputs in programming languages (Rozière et al., 2023; Chen et al., 2021; Li et al., 2023; Gunasekar et al., 2023; Wang et al., 2023; Nijkamp et al., 2023). Complementing the long-standing code generation research (Gulwani et al., 2012; Kurach et al., 2015; Devlin et al., 2017; Parisotto et al., 2016), LLMs can generate programs of more general-purpose programming languages, correctly following programming syntactic rules (Lu et al., 2021; Clement et al., 2020) and solving simple coding problems with reasonable accuracy (Lai et al., 2022; Chen et al., 2021; Austin et al., 2021).

In more direct relevance to our work is the recent line of work for improving code generation qualities through output feedback. Chen et al. (2021) introduced a simple filtering approach by selecting only output samples that successfully pass the public test cases. AlphaCode (Li et al., 2022), CodeT (Chen et al., 2023b), and MBR-Exec (Shi et al., 2022) proposed to generate more test cases and use more sophisticated rule-based methods to rank generation samples by their execution behaviors. LEVER (Ni et al., 2023), Coder-Reviewer (Zhang et al., 2023b) and Code Rankers (Inala et al., 2022) follow a similar principle but introduce more model-based ranking methods.

Recently, more related works have been proposed to boost generation quality through iterative self-revisions. Zhang et al. (2023a) utilizes test outcomes from public test cases as a form of feedback for models to self-revise their codes. Self-correct (Welleck et al., 2023) and CodeRL (Le et al., 2022) introduce secondary models to predict the correctness of output programs and revise them accordingly. Self-debug (Chen et al., 2023c), Sef-refine (Madaan et al., 2023), and Reflexion (Shinn et al., 2023) propose to facilitate better code revision with synthetic natural language explanation or reflection self-generated by LLMs. Self-repair (Olausson et al., 2023) and ILF (Chen et al., 2023a) follow a similar strategy but highlight the use of natural language explanation provided by human experts. Different from prior approaches, we propose to generate more modularized programs and sequentially revise these programs using more representative and reusable sub-module programs (please see Appendix A for a more systematic comparison).

## 3 CODECHAIN FRAMEWORK

### 3.1 CODE GENERATION TASK

We treat code generation as a sequence-to-sequence task, which consists of a problem description as an input sequence $D$ and an output sequence of a flattened solution program: $\hat{W} = (\hat{w}_1, ..., \hat{w}_T)$ with $\hat{w}_t \in \mathcal{V}$. Typically, a language model $\theta$ generates a code sequence by autoregressively sampling

---

**Chain-of-thought prompting for code generation**

*Instruction*
Develop a well-structured Python solution for the provided problem that obeys the constraints and passes the example test cases. Ensure modularity and considering potential edge cases and failures. Start by outlining the required code modules, including function headers and signatures. Subsequently, proceed to implement each module to create the final code.
In simpler terms, create a clean and organized Python solution for the given problem. Break it down into smaller parts (modules) with clear function names and input/output specifications. Once the structure is ready, write the actual code for each module to complete the solution.

---

Figure 3: An example of CoT prompting for code generation in CodeChain. The model is required to first outline the solution in terms of sub-module signatures, each of which is intended for solving a high-level sub-task in the final solution. The model is then required to implement these sub-modules and combine them into a final solution (see Appendix F for a full version of the prompt).

tokens $\hat{w}_t$ from the parameterized conditional distribution $p_\theta(.|\hat{w}_{1:t-1}, D)$. Generated codes are evaluated against (private) test cases to check the execution correctness (Hendrycks et al., 2021; Chen et al., 2021; Li et al., 2022). The test cases comprise a set of input-output pairs $\{(i_j, o_j)\}_{j=1}^{J}$. An output program $\hat{W}$ is correct when $\hat{W}(i_j) = o_j$ for all $j \in \{1, ..., J\}$. If the problem description contains some test cases, we treat these as public test cases: $\{(i'_m, o'_m)\}_{m=1}^{M}$ (usually $M \ll J$). Models have the option to use these public test cases to improve its generation.

## 3.2 MODULAR CODE GENERATION WITH COT PROMPTING

LLMs, especially the instruction-tuned ones, can follow complex natural language instructions describing novel unseen tasks (Ouyang et al., 2022; Touvron et al., 2023b; Wang et al., 2023). They have shown remarkable performance in many reasoning-based tasks when they are instructed to solve a problem step-by-step, i.e., chain-of-thought (CoT) prompting (Zhou et al., 2023; Wei et al., 2022; Kojima et al., 2022). We propose to adapt this technique to generate codes by instructing the models to first outline the required sub-modules, generating only their function headers and docstrings describing their intended usage. The model is then instructed to implement the modules and ultimately combine them into a final solution. Following this generation scheme, we can define the output distributions:

$$\hat{S}_i \sim p_\theta(.|\hat{S}_{1:i-1}, D) \qquad \Rightarrow \text{sub-modules, including the function headers and docstrings} \quad (1)$$

$$\hat{w}_t \sim p_\theta(.|\hat{w}_{1:t-1}, \{\hat{S}_i\}, D) \quad \Rightarrow \text{tokens in final solution} \quad (2)$$

where $\{\hat{S}_i\}$ is the set of sub-modules outlined by the model. We append the instruction with a one-shot demonstration. Figure 3 presents one example of the instruction prompt.

As illustrated further by Figure 10 in the Appendix, this technique encourages the model to decompose a program into natural boundaries, e.g., sub-modules, similarly to how a developer often tackles a challenging coding task by breaking a solution into modular components. Though this is a more pragmatic style of code-development, empirically we have found that this prompting approach can adversely impact the correctness of the generated end-to-end solutions (shown later in Table 3). This is expected as most of the current LLMs are not pretrained to generate perfectly functioning modularized programs. To address this, we introduce Chain of Self-Revisions which allows the LLM to iteratively revise a solution by re-using or adapting some of the representative sub-modules from the previous iterations. Further, we also establish empirically that our self-revision technique indeed benefits more from this modularized style of code generation.

## 3.3 SELECT REPRESENTATIVE SUB-MODULES ACROSS MULTIPLE SAMPLES

Prior studies have demonstrated the benefits of generating multiple samples and selecting the best ones based on different ranking or scoring schemes (Li et al., 2022; Chen et al., 2023b; Zhang et al., 2023b). A common approach is to simply select the representative candidates based on their execution results on the public test cases (Li et al., 2022; Chen et al., 2021). However, all prior methods only select end-to-end program candidates. On challenging coding tasks, it is extremely rare to obtain such

program-level correctness and the selected candidates are still likely to fail when tested on private test cases. Thus, we propose to perform selection at sub-module level instead of program level.

Specifically, given a generation budget of $N$ samples, we extract and combine the set of sub-modules across all samples $\hat{S} = \{\{\hat{S}_i\}_n\}$ for all $n \in \{1, ..., N\}$, where $\{\hat{S}_i\}_n$ is the set of sub-modules in the $n$-th generated sample. We then perform $K$-mean clustering on this set of sub-modules to group them into $K$ clusters. For each of these clusters, we then extract a "centroid" (representative) sub-module $\hat{C}_k$ that is closest to the true centroid of the cluster in the embedding space:

$$\hat{C}_k = \arg\min_{\hat{S}^k} \|\mathcal{S}_i^k - \mu_k\| \tag{3}$$

where $\mathcal{S}_i^k$ is the embedding representation of sub-module $\hat{S}_i$ in cluster $k$ and $\mu_k$ is the centroid of cluster $k$. By selecting these "centroid" sub-modules, we can sample the most semantically representative and re-usable functions across all samples. Note that in cases where public test cases are available, one can filter out any failed samples first before further applying our selection method.

### 3.4 Improve Code Generation with Chain of Self-Revisions

Prior approaches improved code generation by regenerating code conditioned by different types of feedback, ranging from compiler error messages to natural language explanation of the output programs (Chen et al., 2023a; Madaan et al., 2023; Chen et al., 2023c; Shinn et al., 2023; Le et al., 2022). However, these methods focus on the feedback extracted only per individual generation sample.

We propose to utilize a new type of feedback in the form of clustered sub-modules extracted from all the $N$ generated samples (as described in Sec. 3.3). Augmenting our original CoT prompt with the implementations of these representative sub-modules can explicitly encourage the LLM to re-use or adapt these functions when generating the code conditioned on that prompt in the subsequent revision rounds. Specifically, in revision round $R$, the output token is sampled from the conditional distribution:

$$\hat{w}_t^R \sim p_\theta(.|\hat{w}_{1:t-1}^R, \{\hat{S}_i^R\}, \hat{C}^{R-1}, D) \tag{4}$$

where $\hat{C}^{R-1} = \{\hat{C}_k^{R-1}\}_{k=1}^K$ is the set of all centroid sub-modules from the previous generation round $R-1$. In round $R$, the new sub-modules are regenerated by the conditional probability (revised version of Eq. 1):

**Self-revise prompting with representative sub-modules**

*Instruction*
...Given a set of related utility Python functions, try to reuse or adapt them as much as possible into your solution (create new unique functions if needed)....

-----------------
### TASK:
<<problem>>
### RELEVANT FUNCTIONS:
<<sub-modules>>
### RESPONSE:

Figure 4: An example of prompting to self-revise programs. The original instruction from CoT prompting (Fig. 3) is combined with this instruction and the model is provided with a set of representative sub-modules («sub-modules») selected from previously generated samples. Please refer to Appendix F for a full version of the prompt.

$$\hat{S}_i^R \sim p_\theta(.|\hat{S}_{1:i-1}^R, \hat{C}^{R-1}, D) \tag{5}$$

We enable this self-revision procedure by prompting the LLM with an additional instruction. Figure 4 presents an example of the prompt with the new instruction. This style of self-revision with selective sub-modules is reminiscent of the *code reuse* process. In today's agile code-development environment, developers typically re-use or adapt snippets of previously developed code in order to program more modularly, accurately, and efficiently. Inspired by this process and combined with our representative sub-module selection method, our CodeChain framework allows the LLM to iteratively improve their generations more efficiently through a chain of reuse-based self-revisions.

## 4 Experiments

### 4.1 Experimental Setups

**Benchmarks.** We demonstrate the efficacy of CodeChain on challenging code generation tasks, specifically, on two major benchmarks: APPS (Hendrycks et al., 2021), and CodeContests (Li et al.,

Table 1: APPS test results: results with † are for models finetuned on APPS training data

(a) Performance by pass@1 (%)

| Model | Size | Introductory | Interview | Competition | All |
|---|---|---|---|---|---|
| Codex | 12B | 4.14 | 0.14 | 0.02 | 0.92 |
| CodeT5 † | 770M | 6.60 | 1.03 | 0.30 | 2.00 |
| CodeRL+CodeT5 † | 770M | 7.08 | 1.86 | 0.75 | 2.69 |
| text-davinci-002 | - | - | - | - | 7.48 |
| Self-edit+text-davinci-002 | - | - | - | - | 7.94 |
| code-davinci-002 | - | 29.30 | 6.40 | 2.50 | 10.20 |
| WizardCoder | 15B | 26.04 | 4.21 | 0.81 | 7.90 |
| CodeChain+WizardCoder | 15B | 26.29 | 7.49 | 3.75 | 10.50 |
| GPT3.5 | - | 48.00 | 19.42 | 5.42 | 22.33 |
| CodeChain+GPT3.5 | - | **54.50** | **28.11** | **12.38** | **30.24** |

(b) Performance by pass@1 (%) with outputs filtered by public/synthetic tests

| Model | Size | Filtering | Introductory | Interview | Competition | All |
|---|---|---|---|---|---|---|
| Codex | 12B | naive | 22.78 | 2.64 | 3.04 | 6.75 |
| CodeRL+CodeT5 † | 770M | naive | 17.17 | 6.78 | 4.88 | 8.48 |
| code-davinci-002 | - | naive | 43.60 | 13.30 | 7.00 | 18.10 |
| code-davinci-002 | - | CodeT | 47.30 | 14.30 | 6.20 | 19.28 |
| GPT3.5 | - | CodeT | 61.52 | 30.57 | 9.46 | 32.54 |
| CodeChain+GPT3.5 | - | CodeT | **62.72** | **32.96** | **15.08** | **35.34** |

2022). A majority of test samples from these benchmarks are curated from competitive programming platforms such as Codeforces [2], making them an appropriate test bed to evaluate our approach. Please refer to Appendix C and Table 6 for more details of the benchmarks.

**Evaluation.** We followed (Hendrycks et al., 2021; Chen et al., 2021; Li et al., 2022) and evaluated the models using the passing rate metric *pass@k*, defined as the percentage of problems solved by using $k$ generated programs per problem. We focused mainly on *pass@1* in this work and followed (Chen et al., 2021) to calculate the normalized passing rate given a generation budget of $N$ outputs per problem. To apply CodeChain, we fixed the budget in each generation/revision round to $N = 20$ generation samples per problem. After the first round of direct generation, we let the models self-revise generated codes for up to 5 rounds of revision. On APPS and CodeContests, we reported the results on the test split following the best self-revision round performance on the validation set. Across all benchmarks, we fixed the one-shot sample in CoT prompting and revision prompting. We randomly selected this one-shot sample from the APPS training split (see Appendix G).

**Base language models.** We applied CodeChain to both open-sourced and closed-sourced pretrained LLMs, including OpenAI's GPT3.5 and GPT4 (Koubaa, 2023), and WizardCoder (Luo et al., 2023). We evaluated different versions of WizardCoder, with model sizes ranging from 1B to 34B parameters. WizardCoder models are instruction-tuned from strong foundational code LLMs, including StarCoder (Li et al., 2023) and Code LLaMA (Rozière et al., 2023). For OpenAI models, we obtained the generation samples by prompting through the public API access [3]. For WizardCoder, we utilized the HuggingFace-hosted model parameters (Wolf et al., 2019) and vLLM (Kwon et al., 2023) to generate programs. We adopted a default temperature of 0.6 to generate output tokens and a max output length of 2048 tokens. Finally, to fairly compare LLM generation capabilities, we chose to use StarEncoder (Li et al., 2023) to embed sampled sub-modules throughout all experiments.

## 4.2 EXPERIMENTAL RESULTS

**Results on APPS.** We compare our approach with prior LLM baselines like Codex (Chen et al., 2021), CodeT5 (Wang et al., 2021), and code-davinci, as well as code-revision methods such as Self-edit (Zhang et al., 2023a), CodeRL (Wang et al., 2021; Le et al., 2022), and Self-repair (Olausson et al., 2023). Table 1a shows that CodeChain, when applied with base LLMs such as GPT3.5 and WizardCoder 15B, can achieve significant performance gains by the *pass@k*. Specifically, CodeChain

---

[2] https://codeforces.com/

[3] *gpt-3.5-turbo-16k* and *gpt-4* on https://platform.openai.com/docs/models/overview

Table 2: Comparison with Self-repair: following Olausson et al. (2023), we reported the results on the same subset of 20 samples on APPS test split using GPT3.5 and GPT4 as base models. Please refer to Table 5 for the full list of this test subset.

| Model | Feedback source | Introductory | Interview | Competition | All |
|---|---|---|---|---|---|
| Self-repair+GPT4 | GPT4 | 42.64 | 19.33 | 3.67 | 33.30 |
| Self-repair+GPT4 | Human | 62.21 | 45.67 | 14.67 | 52.60 |
| GPT3.5 | - | 30.00 | 18.33 | 0.00 | 23.75 |
| CodeChain+GPT3.5 | Sub-modules | 31.67 | 27.86 | 0.00 | 26.35 |
| GPT4 | - | 42.86 | 18.33 | 13.33 | 34.75 |
| CodeChain+GPT4 | Sub-modules | **71.07** | **55.00** | **23.33** | **61.50** |

Table 3: APPS validation results by *pass@1* (%): we tested CodeChain+GPT3.5 for 1 self-revision round by 3 aspects: prompting, filtering by public tests, and sampling methods for revision (R: random, C: centroid, P: whole programs, and M: sub-modules).

| CoT prompting | filter by public tests | Sampling for revision | Introductory | Interview | Competition | All |
|---|---|---|---|---|---|---|
| - | - | - | 39.00 | 26.50 | 12.50 | 26.00 |
| - | - | R-P | 12.40 | 2.00 | 0.61 | 5.00 |
| - | - | C-P | 23.27 | 9.00 | 3.80 | 12.02 |
| - | ✓ | C-P | 45.20 | 28.03 | 9.80 | 27.68 |
| - | - | - | 33.50 | 23.70 | 10.10 | 22.43 |
| ✓ | - | R-P | 24.40 | 18.80 | 9.20 | 17.47 |
| ✓ | - | C-P | 31.33 | 23.70 | 10.10 | 21.71 |
| ✓ | ✓ | C-P | 45.50 | 33.17 | 11.80 | 30.16 |
| ✓ | ✓ | R-M | 49.30 | 36.90 | 12.40 | 32.87 |
| ✓ | ✓ | C-M | **52.00** | **38.83** | **14.50** | **35.11** |

can achieve $10.50\%$ *pass@1* with WizardCoder as the base model, and $30.24\%$ *pass@1* with OpenAI GPT3.5 as the base model, establishing a new SoTA result on APPS. Previous works (Chen et al., 2021; Li et al., 2022) introduced additional performance results by filtering out generation samples that fail public tests and computed *pass@k* on the filtered set. In this work, we followed the setup proposed by CodeT (Chen et al., 2023b) which utilized more advanced filtering with synthetic test cases (see Appendix F for the prompt we used to generate test cases). Table 1b shows that when evaluated on filtered code samples, our CodeChain+GPT3.5 can achieve SoTA results across all levels of problem difficulty with an average of $35.34\%$ *pass@1*.

From Table 1a, when compared with related approaches such as Self-edit and CodeRL, we observed significant relative performance gains when using CodeChain. In Table 2, following Olausson et al. (2023), to compare with Self-repair, we evaluated our approach over the same test subset of 20 samples (14/3/3 samples of introductory/interview/competition level), using both GPT3.5 and GPT4 as base models. We observed that CodeChain can improve the performance with both base models, with more significant gains using GPT4. Specifically, CodeChain+GPT4 can achieve a SoTA result of $61.50\%$ *pass@1* on average, even outperforming Self-repair+GPT4 with human feedback.

**Analysis on single-round self-revision**. To understand the benefits of CodeChain, we conducted experiments with different variants on the validation split of APPS. Table 3 presents the results on single-round self-revision by 3 main aspects: prompting, filtering by public tests, and sampling methods for conditional revisions. First, we observed that without self-revisions (i.e. direct generation), CoT prompting actually negatively affects the model performance as compared to normal prompting. This observation might be due to the fact that pretrained LLMs are not designed to generate perfectly modularized solutions (they were pretrained on public Github codes without filtering for modularity). However, after applying self-revision, we observe that the modularized approach is better, achieving better performance gains than non-modularized solutions.

Secondly, we found that the best strategy to select representative codes for conditional revision is through clustering. This method can reduce noisy data points and create a better form of feedback to improve the generated codes. Finally, we observed that clustering alone is not sufficient to select the optimal representative samples. Additional filtering by public tests is needed to first shift the output distribution to more likely correct samples before clustering the outputs. To avoid the need for public

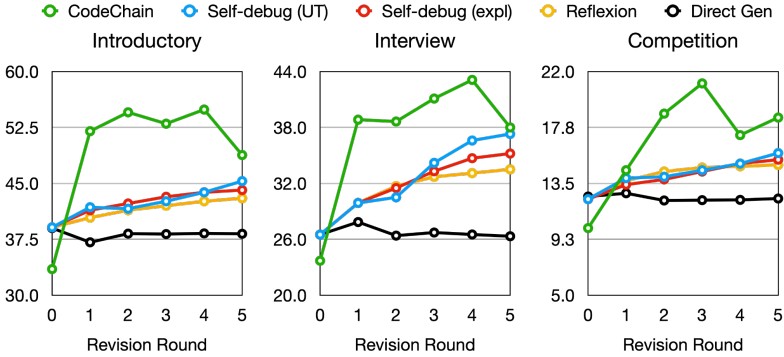
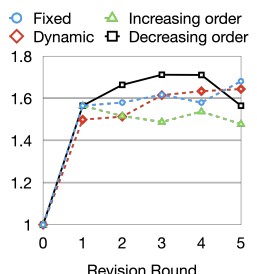

Figure 5: APPS validation results with chain of self-revisions: we tested CodeChain+GPT3.5 for 5 self-revision rounds and reported *pass@1* in each problem difficulty level. Using GPT3.5 as base model, we compared with related approaches, including Self-debug (with unit test (UT) feedback or explanation (expl)) (Chen et al., 2023c) and Reflexion (Shinn et al., 2023).

Figure 6: we tested CodeChain+GPT3.5 on different setups of cluster numbers and reported the average relative *pass@1* improvements from direct generation (round 0).

test cases, we suggest exploring better embedding models that can group output samples not just by their programming semantics but also by their functional correctness.

**Analysis on chain of self-revisions.** To analyze the trend of model performance over a chain of self-revisions, we monitored the passing rates of direct generation and 5 subsequent self-revision rounds. Figure 5 presents relatively consistent improvements in all levels of problem difficulties, with optimal performance gain obtained in revision round 4 and a slight performance drops in round 5. One possible reason for these performance drops is that the selected output samples become overfitting to the small set of available public test cases, negatively affecting the passing rates of subsequently revised codes on a more extensive private hidden test-suite.

Secondly, we also observed that on different levels of problem difficulties, CodeChain has different rates of performance improvement. Specifically, we found that more challenging problems (i.e. competition and interview level) benefit more from CodeChain than basic problems (i.e. introductory level). Similar observations can be seen on open-sourced WizardCoder (Luo et al., 2023), with clearer performance trends on 7B, 15B, and 34B model sizes (see Figure 7).

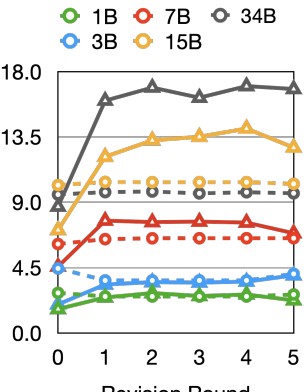

Figure 7: APPS validation *pass@1* results of WizardCoder-1B to 34B. The dotted lines are direct generation results.

**Analysis by types of feedback.** In Figure 5, we also observed that CodeChain can achieve better performance than other related self-revision approaches using other types of feedback, such as test outcomes with natural language explanations (Chen et al., 2023c) or reflection (Shinn et al., 2023). Note that CodeChain can be complemented with other self-revision approaches such as Self-debug by combining different feedback types and selecting more diverse and representative sub-modules, even on generation samples that initially fail public tests.

**Analysis by number of representative sub-modules.** One hyper-parameter of CodeChain is the number of clusters in each round of self-revision. We experimented with 4 different scheme: (i) fixed number of clusters across all rounds to $K$; (ii) decreasing order number of clusters: $\{K_i\} = \{K, K-1, ..., 1\}$; (iii) increasing order number of clusters: $\{K_i\} = \{K, K+1, ...\}$; (iv) dynamic number of clusters based on the silhouette coefficients (Rousseeuw, 1987). We selected $K = 5$ for all experiments. From Figure 6, we observed that the best approach to set the number of clusters is to follow a decreasing order. This scheme offers the models more diverse centroid sub-modules in the beginning with a larger number of clusters. Towards subsequent revision rounds, a smaller number of clusters is more beneficial as the sampled sub-modules become more and more closely semantically similar over time. We found that this scheme is reminiscent of the model training paradigm moving from *exploration* to *exploitation*, as the models become more confident in their generation.

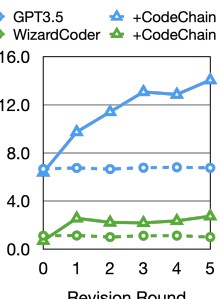

| Model | Size | Filtering | Val | | Test | |
|---|---|---|---|---|---|---|
| | | | pass@1 | pass@5 | pass@1 | pass@5 |
| code-davinci-002 | - | - | - | - | 1.00 | - |
| WizardCoder | 15B | - | 1.11 | 3.18 | 1.98 | 3.27 |
| + CodeChain | 15B | - | 2.35 | 3.29 | 2.48 | 3.30 |
| GPT3.5 | - | - | 6.81 | 16.23 | 5.82 | 11.16 |
| + CodeChain | - | - | **12.86** | **16.91** | **10.27** | **14.11** |
| code-davinci-002 | - | CodeT | - | - | 3.20 | - |
| GPT3.5 | - | CodeT | 17.30 | - | 11.34 | - |
| +CodeChain | - | CodeT | **17.91** | - | **13.75** | - |

Figure 8: CodeContests results by *pass@1* (%): we report the results of CodeChain using WizardCoder-15B and GPT3.5 as base models. Left: test and validation results. Right: validation results over sequential self-revision rounds. The dotted lines are direct generation results.

**Results on CodeContests.** Figure 8 presents the results of CodeChain with WizardCoder-15B and GPT3.5 as the base models. We observed that on both *pass@1* and *pass@5*, CodeChain can achieve significant performance gains as compared to direct generation on the corresponding base models. Applying additional filtering method (Chen et al., 2023b), CodeChain+GPT3.5 can achieve the SoTA results of $13.75\%$ *pass@1* on the test split. As opposed to APPS where optimal performance was reached at revision round $4$, from this validation results we noted that the performance kept improving till the final revision round. Different from APPS, we used the official public test cases available in the CodeContests benchmark. These test cases are generally more diverse than the ones we manually extracted in APPS, and hence, make the revised codes less overfitting even in the $5^{th}$ revision round.

**Qualitative Results.** To understand the modularity and reusability of CodeChain generation, we conducted experiments to evaluate these qualities on randomly sampled generated programs. Specifically, we prompted GPT4 with instructions to rate output samples following a Likert scale from $0$ to $5$ where $5$ is the highest score for optimally modular/ reusable programs. Please refer to Appendix F for a full version of the prompt. In this experiment, we reused the GPT3.5 generated samples for the set of 20 random test tasks from Table 2. Figure 9 shows the distribution of output samples by Likert scores in each quality. We observed that when using CodeChain, GPT3.5 is more likely to generate programs with high levels of modularity and reusability, with the majority of outputs rated 3 to 5 on the Likert scale. This is significantly higher than the conventional direct generation approach, with about $80\%$ of time generating non-modular or non-reusable codes (i.e. score 0). For additional experimental results and qualitative examples of CodeChain, please refer to Appendix D and E.

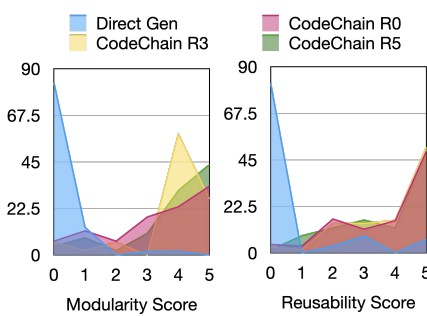

Figure 9: Distribution of output samples (%) by code qualities in the APPS test subset. We obtained the qualitative scores by prompting GPT4 with specific evaluation instructions.

## 5 CONCLUSION

We present CodeChain, a novel inference framework to improve code generation through a chain of self-revisions and sampling of representative sub-modules. In CodeChain, we introduce chain-of-thought prompting to generate more modularized programs, which creates natural boundaries for the models to sample parts of the solutions for reuse and revision. In each revision step, we iterate between selecting representative sub-modules and augmenting chain-of-thought prompting with these selected sub-modules. Our experiments indicate the significant performance improvement of CodeChain when using OpenAI GPT or open-sourced WizardCoder as the base models, achieving new SoTA results on APPS and CodeContests benchmarks. We provided comprehensive ablation studies to understand the contributing factors behind CodeChain's outstanding results.

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

## A  COMPARISON WITH RELATED METHODS

Table 4: A comparison of CodeChain and related approaches by 4 aspects: (i) code execution: whether the method utilizes execution outcomes of output programs on public/synthetic test cases; (ii) representative samples: whether the method sub-samples outputs for evaluation/ revision. (iii) supervision free: whether the method requires model to be finetuned on specialized tasks, such as correctness prediction or bug detection. (iv) iterative revision: whether the method allows model to self-revise output programs multiple times.

| Approach | Code execution | Representative samples | Supervision free | Iterative revision |
|---|---|---|---|---|
| CodeRanker (Inala et al., 2022) | - | ✓ | - | - |
| LEVER (Ni et al., 2023) | ✓ | ✓ | - | - |
| Coder-Reviewer (Zhang et al., 2023b) | - | ✓ | ✓ | - |
| AlphaCode (Li et al., 2022) | ✓ | ✓ | ✓ | - |
| MBR-Exec (Shi et al., 2022) | ✓ | ✓ | ✓ | - |
| CodeT (Chen et al., 2023b) | ✓ | ✓ | ✓ | - |
| Self-correct (Welleck et al., 2023) | - | - | - | ✓ |
| ILF (Chen et al., 2023a) | - | - | - | ✓ |
| CodeRL (Le et al., 2022) | ✓ | - | - | ✓ |
| Self-edit (Zhang et al., 2023a) | ✓ | - | - | ✓ |
| Self-refine (Madaan et al., 2023) | - | - | ✓ | ✓ |
| Self-debug (Chen et al., 2023c) | ✓ | - | ✓ | ✓ |
| Self-repair (Olausson et al., 2023) | ✓ | - | ✓ | ✓ |
| Reflexion (Shinn et al., 2023) | ✓ | - | ✓ | ✓ |
| **CodeChain (ours)** | ✓ | ✓ | ✓ | ✓ |

For a systematic comparison between CodeChain and related approaches, please refer to Table 4.

## B  DEMONSTRATION OF MODULARIZED CODE GENERATION

Please refer to Figure 10 for an illustration of modularized code generation.

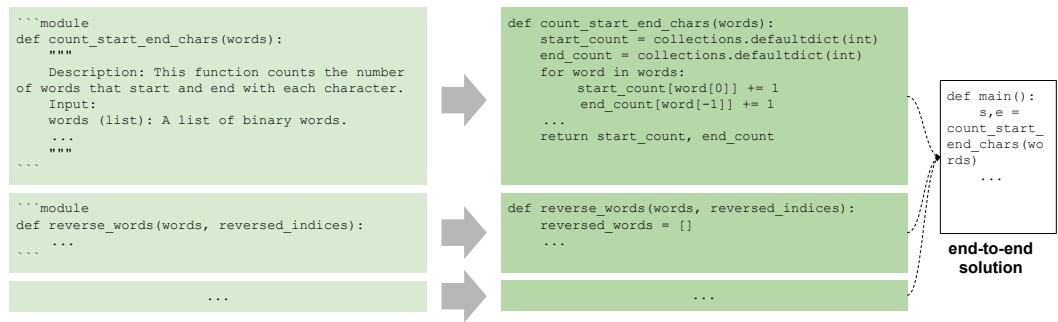

Figure 10: An example of modularized code generation: first, the model is required to outline sub-modules needed, each of which consists of a function header and docstring describing the intended use. Subsequently, the model implements each module fully in code and integrates them as parts of the complete final solution.

## C  ADDITIONAL DETAILS ON BENCHMARKS

Table 6 presents the summary of the benchmarks we used. Note that on APPS, as the original benchmark does not include a specific validation split, we randomly selected samples from the original training split and reported validation results on this set. We sampled 50 samples in each of

Table 5: List of problem ids to create APPS test subset. We followed Olausson et al. (2023) to test on the same subset of 20 samples. In total, we selected 14/3/3 introductory/interview/competition-level samples from the original APPS test split.

| Problem level | Problem IDs |
|---|---|
| Introductory | 4182, 4195, 4281, 4333, 4347, 4426, 4450, 4507, 4514, 4704, 4741, 4855, 4873, 4952 |
| Interview | 2106, 2673, 2923 |
| Competition | 3070, 3286, 3754 |

Table 6: A summary of APPS and CodeContests: † are statistics reported by Li et al. (2022).

| Benchmarks | Val | Test | # Public test cases | # Private test cases |
|---|---|---|---|---|
| APPS (Hendrycks et al., 2021) | 150 | 5000 | 1.98 | 20.99 † |
| CodeContests (Li et al., 2022) | 117 | 165 | 2.00 | 203.70 † |

the 3 levels of difficulty: Introductory, Interview, and Competition. For reproducibility, we included the specific problem ids of this set in Table 7.

We also reported the average number of public and private test cases in each benchmark. Note that on APPS, as the original benchmark did not officially include public test cases, we extracted example input-output pairs from problem descriptions using a rule-based method. We then treated the extracted input-output pairs as public test cases.

## D   ADDITIONAL EXPERIMENTAL RESULTS

**Analysis on chain of self-revisions.** Figure 11 and 12 show the clear performance gains of CodeChain, using GPT3.5 and WizardCoder-34B as the base models, over $5$ rounds of revisions. Specifically on APPS, we found that model performance generally peaks at serf-revision round $4$ (over $1.6x/$ $2x$ performance improvement on average on GPT3.5/ WizardCoder). There is a slight performance drop in round $5$ with GPT3.5 model while the performance is quite stable on WizardCoder. Secondly, we also found that the rates of performance gains are quite different on different levels of problem difficulty. The best improvement from CodeChain is on competition-level problems (with over 2x/ 5x performance gain on GPT3.5/ WizardCoder). This observation indicates that for this type of problem, LLMs can benefit more from leveraging representative modules as a form of hint to revise the generated programs.

In Figure 13, we reported the relative performance gains of CodeChain over multiple rounds of revisions when applying additional filtering on output samples. We observed that compared to direct generation (i.e. round 0), CodeChain can improve the performance of pass@1 by $1.4x$ on filtered outputs, using GPT3.5 as the base model. This observation indicates that CodeChain can complement the line of research works for filtering or sub-sampling output code generations (Chen et al., 2023b; Li et al., 2022; Ni et al., 2023; Shi et al., 2022) by letting the models revise and improve the outputs iteratively.

**Analysis by filtering tests and filtering methods.** To understand the impacts of using test cases for filtering in CodeChain, we conducted ablation experiments with different types of test cases: public/private/synthetic test cases. The synthetic test cases are generated by prompting GPT3.5 with example test cases (see Appendix F for the full version of this prompt). On synthetic test cases, we experimented with sampling generation outputs from the largest clusters (similar to Li et al. (2022)) or by following some consensus of test outcomes among all generation samples (similar to Chen et al. (2023b)).

As can be seen in Table 8, when using synthetic test cases, CodeT sampling approach Chen et al. (2023b) can achieve better performance than other sampling approaches. Both the conventional filtering approach (complete pass) and AlphaCode (sampling from the largest clusters) (Li et al., 2022) led to performance drops, possibly due to the noisy filtered outputs from imperfect synthetic tests. In CodeT, this data noise is addressed through a more sophisticated grouping and sampling method to select better program candidates. However, all results from synthetic tests are not as good

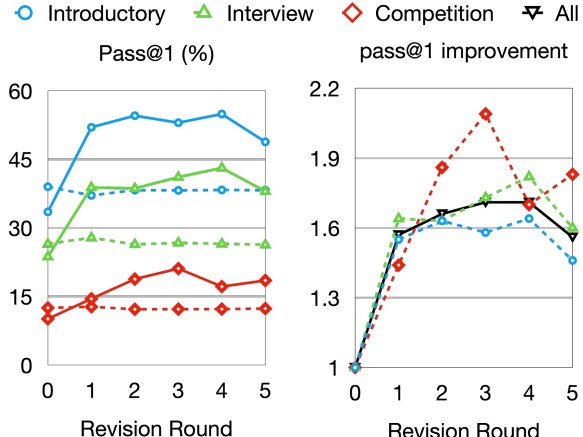

Figure 11: APPS validation results with chain of self-revisions: we tested CodeChain+GPT3.5 for 5 self-revision rounds and reported *pass@1* results. We also reported the relative performance gains from direct generation (i.e. round 0). Note that in the left chart, the dotted lines represent normal direct generation of non-modularized solutions.

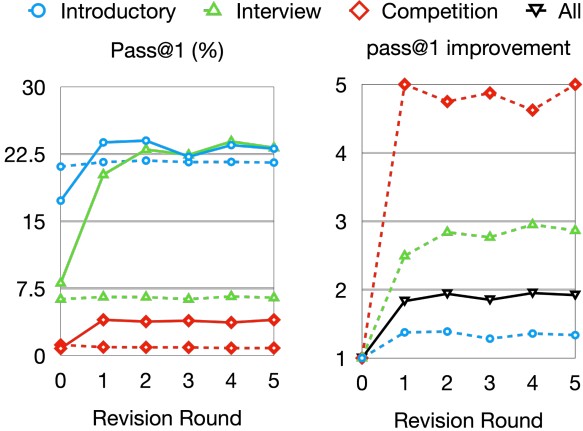

Figure 12: APPS validation results with chain of self-revisions: we tested CodeChain+WizardCoder(34B) for 5 self-revision rounds and reported *pass@1* results. We also reported the relative performance gains from direct generation (i.e. round 0). Note that in the left chart, the dotted lines represent normal direct generation of non-modularized solutions.

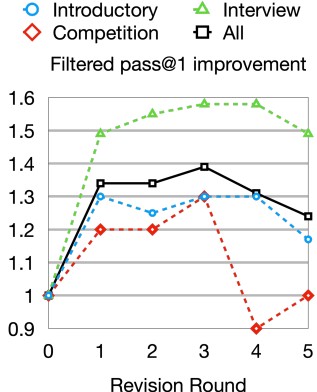

Figure 13: APPS validation filtered *pass@1* results with chain of self-revisions: we report the relative performance gains of CodeChain+GPT3.5 when filtering output samples by synthetic test cases (similarly to CodeT (Chen et al., 2023b)) on all generation/ revision rounds.

Table 7: List of problem ids to create APPS validation split. In total, we selected 50/50/50 introductory/interview/competition-level samples from the original APPS training split.

| Problem level | Problem IDs |
|---|---|
| Introductory | 2361, 2362, 2363, 2364, 2365, 2366, 2367, 2368, 2369, 2370, 2371, 2372, 2373, 2374, 2375, 2376, 2377, 2378, 2379, 2380, 2381, 2382, 2383, 2384, 2385, 2386, 2387, 2389, 2390, 2391, 2392, 2393, 2394, 2395, 2396, 2397, 2398, 2400, 2401, 2402, 2403, 2404, 2405, 2406, 2407, 2408, 2409, 2410, 2411, 2412 |
| Interview | 0, 1, 2, 3, 4, 5, 6, 7, 8, 9, 10, 11, 12, 13, 14, 15, 16, 17, 18, 19, 20, 21, 22, 23, 24, 26, 27, 28, 29, 30, 31, 32, 33, 34, 35, 36, 37, 38, 39, 40, 41, 42, 43, 44, 45, 46, 47, 48, 49, 50 |
| Competition | 2000, 2002, 2003, 2004, 2005, 2006, 2007, 2008, 2010, 2011, 2012, 2013, 2014, 2015, 2016, 2017, 2019, 2020, 2022, 2023, 2024, 2025, 2026, 2027, 2028, 2029, 2030, 2031, 2032, 2033, 2034, 2036, 2037, 2038, 2039, 2040, 2041, 2042, 2043, 2045, 2046, 2048, 2049, 2050, 2051, 2052, 2053, 2054, 2055, 2056 |

Table 8: Ablation results on APPS validation split by *pass@1(%)*: we report the results of CodeChain+GPT3.5 using public/private/synthetic test cases to filter for generation samples before applying grouping the sub-modules into clusters. For synthetic test cases, we experimented with sampling outputs from the largest cluster (following Li et al. (2022)) or sampling outputs by some consensus of test outcomes (following Chen et al. (2023b)).

| Filtering Tests | Filter by | Introductory | Interview | Competition | All |
|---|---|---|---|---|---|
| - | - | 33.50 | 23.70 | 10.10 | 22.43 |
| Synthetic | All Passed | 33.50 (0.0) | 23.70 (0.0) | 10.10 (0.0) | 22.43 (0.0) |
| Synthetic | AlphaCode | 33.15 (-0.4) | 20.98 (-2.7) | 12.00 (+1.9) | 22.04 (-0.4) |
| Synthetic | CodeT | 39.40 (+5.9) | 23.60 (-0.1) | 12.27 (+2.2) | 25.09 (+2.7) |
| Public | All Passed | 52.00 (+18.5) | **38.83 (+15.1)** | 14.50 (+4.4) | 35.11 (+12.7) |
| Private | All Passed | **55.70 (+22.2)** | 37.60 (+13.9) | **18.90 (+8.8)** | **37.40 (+15.0)** |

as ones with public test cases available in problem descriptions. This observation indicates that CodeChain is quite sensitive to the correctness filtering and has to rely on high-quality test cases. Finally, as expected, we found that the best performance is obtained by filtering against private test cases which can cover more corner cases and lead to better filtered programs for self-revisions.

**Analysis by different embedding models.** Table 9 presents the results of CodeChain+GPT3.5 when using different embedding methods: StarCoder (Li et al., 2023), CodeT5+ (Wang et al., 2023), and CodeBert (Feng et al., 2020). We found that CodeT5+ can outperform CodeBert in all levels of problem difficulty, especially on the introductory-level problems. Among all embedding methods, using StarCoder with CodeChain can achieve the best performance overall. We noted that one major advantage of StarCoder is the ability to encode long-context programs, which we found quite frequently in many challenging coding problems in APPS and CodeContests.

**Using public tests as a proxy to select optimal revision step.** In all our experiments, we selected revised generation samples during test time on the best revision round we found by validation results. However, in practice, we propose to use the public tests as a proxy to gauge model performance throughout the chain of self-revisions. Figure 14 shows that on APPS, performance on public tests is often correlated well with the performance on private test cases. Specifically, the model peaked consistently at revision round 4 in both sets of test cases. Therefore, in practice, even without a validation set, we can still apply CodeChain using public test results as a stopping signal to select the best version of revised programs.

**Results on LeetCodeHardGym.** Finally, we attempted to apply CodeChain on the recent code generation benchmark LeetCodehardGym (Shinn et al., 2023), including a set of 39 challenging coding tasks extracted from LeetCode. Figure 15 shows that GPT4+CodeChain can significantly improve the performance of using GPT4 alone. Different from APPS, we observed that the model achieves the best performance at revision round 3, achieving close to 7% *pass@1*. Note that our experimental setup here is quite different from the original setup in (Shinn et al., 2023). We could only test on 39/40 original test problems as 1 problem was withdrawn from the released benchmark due to erroneous test data. Also, in (Shinn et al., 2023), the public test suite was synthetically created by LLMs and validated by AST. In our setup, we simply used the available public tests extracted

Table 9: Ablation results on APPS validation split by *pass@1(%)*: we report the results of CodeChain+GPT3.5 when clustering sub-modules by different embedding methods.

| Embedding Model | Introductory | Interview | Competition | All |
|---|---|---|---|---|
| - | 33.50 | 23.70 | 10.10 | 22.43 |
| CodeBert | 50.73 (+17.2) | 34.80 (+11.1) | 13.80 (+3.7) | 33.11 (+10.7) |
| CodeT5+ | 53.00 (+19.5) | 35.37 (+11.7) | 13.10 (+3.0) | 33.82 (+11.4) |
| StarCoder | 52.00 (+18.5) | 38.83 (+15.1) | 14.50 (+4.4) | 35.11 (+12.7) |

from the problem descriptions. Nevertheless, we still observed consistent benefits of CodeChain in this benchmark, complementing other related results of CodeChain in APPS and CodeContests.

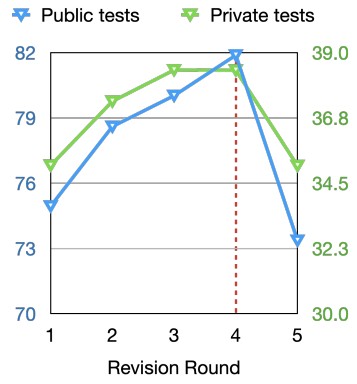

Figure 14: APPS validation results by pass@1 (%) on public vs. private test cases

Figure 15: LeetCodeHardGym *pass@1* results using GPT4+CodeChain on 5 revision rounds.

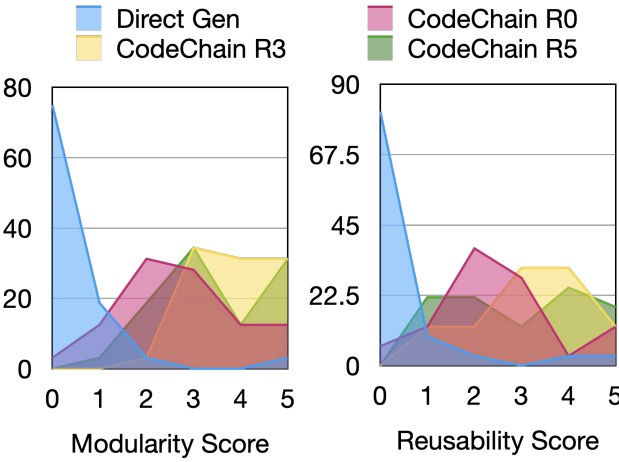

Figure 16: Distribution of output samples (%) by code qualities in the APPS test subset: we obtained the qualitative scores by conducting a survey with human judges to evaluate the qualities of generation samples of CodeChain (revision round 0 to 5) and direct generation. Please refer to Figure 17 and 18 for a copy of the survey guideline and an example survey question.

**Human evaluation results.** We conducted a survey with human judges to estimate how generation samples from CodeChain aligns with human preference by two qualities: reusability and modularity. We referred to the set of instruction we used to prompt GPT-4 for automatic evaluation and reused them to create the survey guideline. The human judges are required to follow a Likert scale from 0 to 5 where 5 is the highest score for optimally modularized/ resuable programs. We reused the same subset of samples from GPT-4 evaluation (20 random test tasks from Table 2) so that we can fairly compare with GPT-4 based results. Figure 16 shows the distribution of output samples by

Likert scores in each quality. We observed that compared to the GPT-4 based results (Figure 9), the human evaluation results of CodeChain are slightly different with more even distributions across the upper range (score $3 - 5$ in both modularity and reusability). However, we still observed a consistent improvement when comparing CodeChain against the traditional direct generation approach. Please refer to Figure 17 and 18 for a copy of the survey instruction/guideline to human judges and an example survey question.

## Final Evaluation for CodeChain

**Judging Modularity**:
A code is modular if it is decomposed into logical atomic sub-modules each handling different sub-tasks of
the given task. Sub-modules are functions that are invoked from the 'main' method.
Based on your judgement give a rating on a Likert Scale from [0-5] to the given solution.
**Modularity Ratings**:
- Rating 5: Optimally Modular - If the candidate solution is optimally modular
- Rating 1 to 4: Less Modular - If the solution could have been more modular or if the sub-modules are not
atomic enough
- Rating 0: Not Modular - If the solution is flat block of code without any decomposition into
 sub-modules

**Judging Re-usability**:
A sub-module is re-usable if it is generic enough that its implementation can be re-used as-is for other
problems for this domain. If they have any specific characteristic or part of the implementation that
is only particular to this given task and is less likely to appear in other tasks, then the sub-module
is not re-usable.
Only if the candidate solution is at least somewhat modular and there exists at least one sub-module, for
each sub-module give a rating on a Likert Scale from [0-5] its re-usability based on the above
judgement. If you rate it less than 5, provide the justification.
**Re-usability Ratings**:
- Rating 5: the function is generic and reusable in other problems
- Rating 1 to 4: the function is somewhat generic and might be reusable only in problems quite similar to the
given task
- Rating 0: the function is not at all generic or reusable in other problems
If the candidate solution is not modular such reusability rating is not needed.

Last, note that 'main' function is not a sub-module.

Figure 17: Survey guideline for human evaluation: we adapted the instructions we used for GPT4-based evaluation to create a survey guideline for our human evaluation experiments. For each generation sample (either from CodeChain or direct generation baseline), human judges are required to give rate it from 0 to 5 on 2 specific code qualities: modularity and reusability.

## E    EXAMPLE GENERATION SAMPLES

For the same input problem, we presented different generation samples generated by GPT4 and CodeChain+GPT4. In each generation sample, we appended the evaluation generated by GPT4 regarding the modularity and reusability of the output program. Through automatic evaluation

```
 1  def read_input():
 2      x1, y1, x2, y2 = map(int,input().split())
 3      return x1, y1, x2, y2
 4
 5  def calculate_square_vertices(x1, y1, x2, y2):
 6      x3 = x2 - (y2 - y1)
 7      y3 = y2 + (x2 - x1)
 8      x4 = x1 - (y2 - y1)
 9      y4 = y1 + (x2 - x1)
10      return x3, y3, x4, y4
11
12  def print_output(x3, y3, x4, y4):
13      print(x3, y3, x4, y4)
14
15  def solve_square():
16      x1, y1, x2, y2 = read_input()
17      x3, y3, x4, y4 = calculate_square_vertices(x1, y1, x2, y2)
18      print_output(x3, y3, x4, y4)
19
20  solve_square()
```

|             | 0 | 1 | 2 | 3 | 4 | 5 |
|-------------|---|---|---|---|---|---|
| Modularity  | ○ | ○ | ○ | ○ | ○ | ○ |
| Re-usability| ○ | ○ | ○ | ○ | ○ | ○ |

Figure 18: An example survey question with a generation sample and the choices a human judge can choose to rate the code (0 to 5) by its modularity and reusability qualities.

metrics, we found that the generation samples gradually became more modularized following the chain of revisions. In general, the generation samples are also more reusable as compared to the normal direct generation approach (without CodeChain). Through manual inspection, we observed that CodeChain can indeed generate better programs, properly modularized and well explained with function docstrings.

### E.1    EXAMPLE OF INTRODUCTORY-LEVEL PROBLEM AND CODECHAIN GENERATION SAMPLES

Listing 1: An example of problem description from APPS (problem ID 4195)

```
Today, the memorable AtCoder Beginner Contest 100 takes place. On this occasion, Takahashi would like to give
     an integer to Ringo.

As the name of the contest is AtCoder Beginner Contest 100, Ringo would be happy if he is given a positive
     integer that can be divided by 100 exactly D times.
Find the N-th smallest integer that would make Ringo happy.

-----Constraints-----
 - D is 0, 1 or 2.
 - N is an integer between 1 and 100 (inclusive).

-----Input-----
Input is given from Standard Input in the following format:
D N

-----Output-----
Print the N-th smallest integer that can be divided by 100 exactly D times.

-----Sample Input-----
0 5

-----Sample Output-----
5

The integers that can be divided by 100 exactly 0 times (that is, not divisible by 100) are as follows: 1, 2,
     3, 4, 5, 6, 7, ...

Thus, the 5-th smallest integer that would make Ringo happy is 5.
```

Listing 2: An example of output generated by direct generation with GPT4

```python
```

```
D, N = map(int, input().split())
if N == 100:
    N += 1
print(100**D * N)
```
This problem can be solved by simple math. In Python, ** is used to calculate the power of a number.
    Therefore, 100 to the power of D is the factor by which we multiply N. However, if N is 100, we count
    101 as the 100th number because 100 itself is excluded as it can be divided by 100 but not exactly D
    times.

==================================================
### OpenAI evaluation of the above output by modularity and reusability
### Modularity
0

### Re-usability
The code is not modular. So, re-usability rating is not applicable.
```

Listing 3: An example of output generated by GPT4 with CoT prompting (revision round 0 of CodeChain)

```
STEP 1: GENERATE SUB-MODULES
```module
def Nth_smallest_integer(D, N):
    '''
    This function finds the nth smallest integer that can be divided by 100 exactly D times.

    D (int): a positive integer 0, 1 or 2.
    N (int): a positive integer between 1 and 100

    Output:
        result: the N-th smallest integer that can be divided by 100 exactly D times.
    '''
```

STEP 2: GENERATE PYTHON CODE
```python
def Nth_smallest_integer(D, N):
    if N == 100:
        N += 1
    result = (100 ** D) * N
    return result

D, N = map(int, input().split())
print(Nth_smallest_integer(D, N))
```

This code works by multiplying 100 raised to the power of D times with input number N. If N is 100, increment
     N by 1 since it needs to be the next contender for 100's divisible number.

==================================================
### OpenAI evaluation of the above output by modularity and reusability
### Modularity
2

### Re-usability
- nth_smallest_integer: 3. Justification: The function is specific to the task at hand and not generic enough
      to be reused in other problems. While the principle of multiplying by a certain power of a number could
      potentially be applied in other problems, the hardcoded values and the specific edge-case makes it less
      likely to be directly reusable.
```

Listing 4: An example of output generated by GPT4 with CodeChain self-revision (revision round 3)

```
STEP 1: GENERATE SUB-MODULES:

```module
def get_input():
    """
    Description: This function gets inputs from the user/stdin.
    Input: None
    Output:
    D (int): Divisibility with 100 factor
    N (int): The Nth term needed
    """
```

```module
def happy_number(D, N):
    """
    Description: Calculate the Nth happy number according to the given divisibility rule.
    Input:
    D (int): Divisibility with 100 factor
    N (int): The Nth term needed
    Output:
    int: the N-th smallest integer that can be divided by 100 exactly D times.
    """
```
```

```module
def print_output(number):
    """
    Description: This function prints the answer.
    Input:
    number (int): the corresponding number.
    Output:
    None
    """
```

STEP 2: GENERATE PYTHON CODE

```python
def get_input():
    D, N = map(int, input().split())
    return (D, N)

def happy_number(D, N):
    if N == 100:
        N += 1
    return 100**D * N

def print_output(number):
    print(number)

D, N = get_input()
number = happy_number(D, N)
print_output(number)
```

==================================================
### OpenAI evaluation of the above output by modularity and reusability
### Modularity
5

### Re-usability
- get_input: 3 (Justification: This function is somewhat generic and it might be used in other places that
    require the re-use of two space-separated integer inputs. However, the number of inputs and the type is
    fixed hence not very re-usable. )
- happy_number: 2 (Justification: This function is less generic and it is specific to solve the
    "happy_number" problem due to its specific logic that won't be encountered in most other problem
    scenarios.)
- print_output: 4 (Justification: This function is semi-generic, you can print any number with it, but it is
    limited to integers only, if we have other data types we could not be able to utilise this function
    appropriately.)

Listing 5: An example of output generated by GPT4 with CodeChain self-revision (revision round 5)

```
STEP 1: GENERATE SUB-MODULES:
```module
def get_input():
    """
    Description: This function gets inputs from the user/stdin.
    Input: None
    Output:
    D (int): Divisibility with 100 factor
    N (int): The Nth term needed
    """
```

```module
def print_output(number):
    """
    Description: This function prints the answer.
    Input:
    number (int): the currespond number.
    Output:
    None
    """

    print(number)
```

```module
def find_nth_integer(D, N):
    """
    Description: This function computes the N-th smallest integer that can be divided by 100 exactly D times.
    Input:
    D (int): Divisibility with 100 factor
    N (int): The Nth term needed
    Output:
    number (int): the number that would make Ringo happy
    """
```

STEP 2: GENERATE PYTHON CODE

```python
def get_input():
```

```
    D, N = map(int, input().split())
    return (D, N)

def print_output(number):
    print(number)

def find_nth_integer(D, N):
    if N == 100:
        N += 1
    return 100 ** D * N

D, N = get_input()
number = find_nth_integer(D, N)
print_output(number)
```

==================================================
### OpenAI evaluation of the above output by modularity and reusability
ROUND 5
### Modularity
5

### Re-usability
- get_input: 5
- print_output: 5
- find_n_smallest_integer: 1. Justification: This function is very specific to the problem at hand and needs
      to meet very specific conditions, thus its reusability is limited. However, it could be used in similar
      problems with slight modifications.
```

## E.2 EXAMPLE OF COMPETITION-LEVEL PROBLEM AND CODECHAIN GENERATION SAMPLES

### Listing 6: An example of problem description from APPS (problem ID 3286)

```
Your friend Edvin loves languages. A personal favourite of his is speaking the Robber Language. For those of
      you not familiar with it, it is a language transformation where every consonant is doubled, with an ''o
      put in-between. The rest is kept the same. For example, the word "car" would be transformed into
      "cocaror".

Your friend Edvin was going to tell you his password to his favourite website, so that you could check out
      how awesome it is. Edvin is sure that nobody but him and you speak the Robber Language, so he encrypted
      it using this simple technique, and wrote it down on a note. After trying the password and realising it
      does not work, you found out that Edvin was drunk during the transformation. Now you want to find out
      in how many ways the password might have looked originally.

You are given an encrypted password which Edvin has attempted to translate into the Robber Language. Your
      task is to determine how many passwords might produce the encrypted password. You may assume that the
      only thing Edvin did wrong, was when scanning through the word with his eyes, he sometimes missed to
      transform some consonants using the rule above. Edvin is not deterministic in his failures, i.e. if he
      missed to transform a consonant once, it is not necessary that he would miss it if it occured again
      later in the string.

A vowel is considered to be one of the letters "a", "e", "i", "o" and "u". A consonant is any other letter.

-----Input-----
Input consists of one encrypted password, the resulting word from Edvins translation. The word can be as long
      as $1000000$ characters. It will solely consist of lower case letters a-z.

-----Output-----
Output should consist of one number, the number of ways the password might have looked before Edvin
      translated it. Since this number may be huge, print the remainder modulo $1000009$.

-----Examples-----
Sample Input 1:
car
Sample Output 1:
1

Sample Input 2:
cocar
Sample Output 2:
2
```

### Listing 7: An example of output generated by direct generation with GPT4

```python
s = input()
n = len(s)
vow = set(["a", "e", "i", "o", "u"])
dp = [0]*(n+1)
mod = 10**6+9

dp[0] = 1
dp[1] = 1

for i in range(2, n + 1):
    if s[i-1] in vow or s[i-2] in vow or i - 2 < 0 or s[i-1] != s[i-2] or s[i-1] == 'o':
        dp[i] = dp[i-1]
```

```
    else:
        dp[i] = dp[i-1] + dp[i-2]
        dp[i] %= mod

print(dp[n])
```

==================================================
### OpenAI evaluation of the above output by modularity and reusability
### Modularity
0

### Re-usability
The solution is not modular, hence re-usability rating is not applicable.
```

**Listing 8:** An example of output generated by GPT4 with CoT prompting (revision round 0 of CodeChain)

```
STEP 1: GENERATE SUB-MODULES:

```module
def is_vowel(ch):
    """
    Description: This function returns True if the specified character is a vowel, False otherwise.
    Input:
    ch (char): A character.
    Output:
    (bool): True if the character is a vowel, False otherwise.
    """

def count_possible_original_words(word):
    """
    Description: This function counts the possible original words that could have transformed into the given
     Robber Language word.
    Input:
    word (str): A Robber Language word.
    Output:
    count (int): A count of possible original words.
    """
```

STEP 2: GENERATE PYTHON CODE

```python
MOD = 1000009

def is_vowel(ch):
    vowels = ['a', 'e', 'i', 'o', 'u']
    return ch in vowels

def count_possible_original_words(word):
    N = len(word)
    dp = [0]*(N+1)
    dp[0] = 1
    for i in range(1, N+1):
        dp[i] = dp[i-1]
        if not is_vowel(word[i-1]) and i >= 2 and word[i-1] == word[i-2] and word[i-3] != 'o':
            dp[i] += dp[i-2]
        dp[i] %= MOD
    return dp[N]

word = input().strip()
print(count_possible_original_words(word))
```
This python function works as follows:
It counts the number of ways we can form the Robber language word by taking a dynamic programming approach.
    The `dp` list is initialized with zeroes with `dp[0]` set to `1`. Then for each character in the word,
    it checks whether it is a consonant and matches the preceding character and the character before that
    is not 'o'. If these conditions are met, it counts this as a possible place where a consonant could
    have been skipped and adds `dp[i-2]` to `dp[i]`. Finally, it returns `dp[N]` which contains the total
    number of possible original words. The modulo operation is used to keep the result from overflowing the
    32-bit integer limit.

==================================================
### OpenAI evaluation of the above output by modularity and reusability
### Modularity
Rating: 0

### Re-usability
Since there are no sub-modules, there is no re-usability to rate.
```

**Listing 9:** An example of output generated by GPT4 with CodeChain self-revision (revision round 3)

```
STEP 1: GENERATE SUB-MODULES:

Since we're given two very relevant functions "split_word" and "count_possible_passwords", we can keep these
    functions in our solution. The function "initialize_vowels_and_modulo" can be useful to initialize
    global variables.
```

````
```module
def split_word(word):
    """
    Description: This function splits the word into a list of elements that can be translated back into
     Robber language.
    Input:
    word (str): The word that needs to be split.
    Output:
    split_word_list (list): The list of strings, where each string represents a possible original letter
    """
```

```module
def initialize_vowels_and_modulo():
    """
    Description: This function initializes the vowels and modulo variables.
    Output:
    vowels (set): A set of vowels
    modulo (int): The modulo constant
    """
```

```module
def count_possible_passwords(word_list):
    """
    Description: This function counts the number of possible password that might have produced the original
     word.
    Input:
    word_list (list): A list with elements from the word
    Output:
    (int): The number of possible passwords modulo 1000009
    """
```

STEP 2: GENERATE PYTHON CODE

```python
def split_word(word):
    i = 0
    split_word_list = []
    while i < len(word):
        if i + 2 < len(word) and word[i + 1] == 'o' and word[i + 2] != 'o':
            split_word_list.append(word[i:i+3])
            i += 3
        else:
            split_word_list.append(word[i])
            i += 1
    return split_word_list

def count_possible_passwords(word_list, vowels, modulo):
    dp = [0] * len(word_list)
    dp[0] = 1 if len(word_list[0]) == 1 or word_list[0][0] in vowels else 2
    for i in range(1, len(word_list)):
        if len(word_list[i]) == 1 or word_list[i][0] in vowels:
            dp[i] = dp[i - 1]
        else:
            dp[i] = (dp[i - 1] * 2) % modulo
    return dp[-1]

def initialize_vowels_and_modulo():
    return set(['a', 'e', 'i', 'o', 'u']), 1000009

def solve_task(word):
    split_word_list = split_word(word)
    vowels, modulo = initialize_vowels_and_modulo()
    return count_possible_passwords(split_word_list, vowels, modulo)

word = input()
print(solve_task(word))
```
````

In the final code, function "solve_task()" uses "split_word()" to break down the word into possible original
    letters, "initialize_vowels_and_modulo" to get the vowel set and modulo values. Finally it uses
    "count_possible_passwords()" to calculate the number of possible original passwords.

==================================================
### OpenAI evaluation of the above output by modularity and reusability
### Modularity
4

### Re-usability
- split_word: 5
- initialize_vowels_and_modulo: 3. Justification: Functions setting constants can be re-used, but the
    function name and content are very specific to the task.
- count_possible_passwords: 2. Justification: It is specific to current problem, as it is specific to 'Robber
    Language' problem.
- solve_task: 2. Justification: This function is overall problem specific not re-usable for other problems.

Listing 10: An example of output generated by GPT4 with CodeChain self-revision (revision round 5)

```
STEP 1: GENERATE SUB-MODULES:
```module
def is_vowel(ch):
    """
    Description: This function checks whether a character is a vowel.
    Input:
    ch (str): A single character string.
    Output:
    (bool): Returns 'True' if the character is a vowel, 'False' otherwise.
    """
```

```module
def split_word(word):
    """
    Description: This function splits the word into a list of elements that can be translated into the
      original letters.
    Input:
    word (str): The encrypted password.
    Output:
    split_word_list (list): A list of strings, where each string represents a possible original letter.
    """
```

```module
def count_possibilities(split_word_list):
    """
    Description: This function counts the number of ways each element in the list could be translated into
      the original letters.
    Input:
    split_word_list (list): A list of strings, where each string is an element that could possibly be
      translated into an original letter.
    Output:
    possibilities (int): The total number of ways the password might have looked originally.
    """
```

STEP 2: GENERATE PYTHON CODE
```python
def is_vowel(ch):
    return ch in 'aeiou'

def split_word(word):
    i = 0
    split_word_list = []
    while i < len(word):
        if i + 2 < len(word) and not is_vowel(word[i]) and word[i + 1] == 'o' and word[i] == word[i + 2]:
            split_word_list.append(word[i:i+3])
            i += 3
        else:
            split_word_list.append(word[i])
            i += 1
    return split_word_list

def count_possibilities(split_word_list):
    possibilities = 1
    modulo = 1000009
    for element in split_word_list:
        if len(element) == 1 and not is_vowel(element):
            possibilities *= 2
            possibilities %= modulo
    return possibilities

word = input()
split_word_list = split_word(word)
possibilities = count_possibilities(split_word_list)
print(possibilities)
```
This solution makes use of the 'split_word' and 'is_vowel' relevant functions. The 'count_possibilities'
    function is new and quite specific to this problem. In 'split_word', we divide the word into potential
    original words assuming that Edvin didn't make any mistake when he processed a consonant. Then, if it's
    a single character and is a consonant, it means it could either have been an original consonant that
    Edvin didn't process or a consonant that Edvin processed correctly.

=================================================
### OpenAI evaluation of the above output by modularity and reusability
### Modularity
5

### Re-usability
- is_consonant: 5, Justification: A function to check if a given character is a consonant or not is generic
    and can be used in a wide variety of tasks involving character manipulation or language processing.
- split_word: 3, Justification: This function is less generic than 'is_consonant' because it is specifically
    designed to split a string in the special way that is relevant for the Robber Language. However, it
    might be useful in similar tasks involving specially formatted string decoding.
```

```
- solve_task: 1, Justification: This function is very specific to the current task and is unlikely to be
      reused in other problems as it implements the logic for deriving possible original passwords from the
      encrypted password in Robber language.
```

## F  PROMPTS WITH INSTRUCTION

Listing 11: CoT prompting with instruction to generate modularized code. «question_guide» is replaced with instructions for the model to follow either standard input streams or call-based functions.

```
*Instruction*
Develop a well-structured Python solution for the provided problem that obeys the constraints and passes the
      example test cases. Ensure modularity and considering potential edge cases and failures. Start by
      outlining the required code modules, including function headers and signatures. Subsequently, proceed
      to implement each module to create the final code.

In simpler terms, create a clean and organized Python solution for the given problem. Break it down into
      smaller parts (modules) with clear function names and input/output specifications. Once the structure
      is ready, write the actual code for each module to complete the solution.
The output code needs to <<question_guide>>. Please wrap your code answer using '''.
### Example 1
### TASK:
<<example problem>>
### RESPONSE:
STEP 1: GENERATE SUB-MODULES:
<<example generated sub-modules>>
STEP 2: GENERATE PYTHON CODE
<<example generated code>>
----------------
### Example 2
### TASK:
<<new problem>>
### RESPONSE:
```

Listing 12: self-revision prompting with instruction to revise and generate modularized code. «question_guide» is replaced with instructions for the model to follow either standard input streams or call-based functions. «sub_modules» is replaced with representative sub-modules selected by CodeChain framework.

```
*Instruction*
Develop a well-structured Python solution for the provided problem that obeys the constraints and passes the
      example test cases. Ensure modularity and considering potential edge cases and failures. Given a set of
      related utility Python functions, try to reuse or adapt them as much as possible into your solution
      (create new unique functions if needed). Start by outlining the required code modules, including
      function headers and signatures. Subsequently, proceed to implement each module to create the final
      code.

In simpler terms, create a clean and organized Python solution for the given problem. Break it down into
      smaller parts (modules) with clear function names and input/output specifications. Once the structure
      is ready, write the actual code for each module to complete the solution.

The output code needs to <<question_guide>>. Please wrap your code answer using '''.
### Example 1
### TASK:
<<example problem>>
### RELEVANT FUNCTIONS:
<<sub-modules>>
### RESPONSE:
STEP 1: GENERATE SUB-MODULES:
<<example generated sub-modules>>
STEP 2: GENERATE PYTHON CODE
<<example generated code>>
----------------
### Example 2
### TASK:
<<new problem>>
### RELEVANT FUNCTIONS:
<<sub-modules>>
### RESPONSE:
```

Listing 13: prompt to generate synthetic test cases to use as additional public tests. «example_test» is replaced with any available test cases extracted in the problem description. We expect the model to follow similar formats to the example test cases and continue to generate up to 20 test cases in total.

```
### Question:
Generate 20 new input and output pairs as example test cases for the following problem:

<<problem>>

Please response following this format:

## Test 1:
```

```
Input:
<input>

Output:
<expected output>

## Test 2:
...

### Answer:

20 test cases:

<<example_test>>
```

Listing 14: prompt to evaluate code generation samples by their modularity and reusability qualities.

```
You are an expert software engineer and you always emphasize on writing modular reusable codes, for any given
    task description.
You are given a task description and a candidate solution in form of a python code for it. You have to judge
    the following aspects of the code:

### Judging Modularity:
A code is modular if it is decomposed into logical atomic sub-modules each handling different sub-tasks of
    the given task. Sub-modules are functions that are invoked from the 'main' method.
Based on your judgement give a rating on a Likert Scale from [0-5] to the given solution.

Modularity Ratings:
- Rating 5: Optimally Modular - If the candidate solution is optimally modular
- Rating 1 to 4: Less Modular - If the solution could have been more modular or if the sub-modules are not
    atomic enough
- Rating 0: Not Modular - If the solution is flat block of code without any decomposition into sub-modules

### Judging Re-usability:
A sub-module is re-usable if it is generic enough that its implementation can be re-used as-is for other
    problems for this domain. If they have any specific characteristic or part of the implementation that
    is only particular to this given task and is less likely to appear in other tasks, then the sub-module
    is not re-usable.
Only if the candidate solution is at least somewhat modular and there exists at least one sub-module, for
    each sub-module give a rating on a Likert Scale from [0-5] its re-usability based on the above
    judgement. If you rate it less than 5, provide the justification.

Re-usability Ratings:
- Rating 5: the function is generic and reusable in other problems
- Rating 1 to 4: the function is somewhat generic and might be reusable only in problems quite similar to the
    given task
- Rating 0: the function is not at all generic or reusable in other problems

If the candidate solution is not modular such reusability rating is not needed.
Also, 'main' function is not a sub-module.

Your response should only be the modularity and re-usability ratings. Judging the correctness of the
    candidate solution or implementing a correct solution for the task is not needed.

Your overall response should follow the format:
### Modularity
0-5 rating

### Re-usability
- <sub-module1>: <rating:0-5 rating. Justification: Justification if not-reusable. >
- <sub-module2>: <rating ...>
- <sub-module3>: <rating ...>
- ... ...

### TASK:
<<TASK>>

### CANDIDATE SOLUTION:
<>

### RESPONSE:
```

# G   ONE-SHOT EXAMPLES

Listing 15: one-shot example input for normal code generation prompting or prompting to generate modularized solutions

```
### Example 1

Polycarp has $n$ different binary words. A word called binary if it contains only characters '0' and '1'. For
    example, these words are binary: "0001", "11", "0" and "0011100".

Polycarp wants to offer his set of $n$ binary words to play a game "words". In this game, players name words
    and each next word (starting from the second) must start with the last character of the previous word.
    The first word can be any. For example, these sequence of words can be named during the game: "0101",
    "1", "10", "00", "00001".
```

```
Word reversal is the operation of reversing the order of the characters. For example, the word "0111" after
    the reversal becomes "1110", the word "11010" after the reversal becomes "01011".

Probably, Polycarp has such a set of words that there is no way to put them in the order correspondent to the
    game rules. In this situation, he wants to reverse some words from his set so that:  the final set of
    $n$ words still contains different words (i.e. all words are unique);  there is a way to put all words
    of the final set of words in the order so that the final sequence of $n$ words is consistent with the
    game rules.

Polycarp wants to reverse minimal number of words. Please, help him.

-----Input-----

The first line of the input contains one integer $t$ ($1 \le t \le 10^4$) -- the number of test cases in the
    input. Then $t$ test cases follow.

The first line of a test case contains one integer $n$ ($1 \le n \le 2\cdot10^5$) -- the number of words in
    the Polycarp's set. Next $n$ lines contain these words. All of $n$ words aren't empty and contains only
    characters '0' and '1'. The sum of word lengths doesn't exceed $4\cdot10^6$. All words are different.

Guaranteed, that the sum of $n$ for all test cases in the input doesn't exceed $2\cdot10^5$. Also, guaranteed
    that the sum of word lengths for all test cases in the input doesn't exceed $4\cdot10^6$.

-----Output-----

Print answer for all of $t$ test cases in the order they appear.

If there is no answer for the test case, print -1. Otherwise, the first line of the output should contain $k$
    ($0 \le k \le n$) -- the minimal number of words in the set which should be reversed. The second line
    of the output should contain $k$ distinct integers -- the indexes of the words in the set which should
    be reversed. Words are numerated from $1$ to $n$ in the order they appear. If $k=0$ you can skip this
    line (or you can print an empty line). If there are many answers you can print any of them.

-----Example-----
Input
4
4
0001
1000
0011
0111
3
010
101
0
2
00000
00001
4
01
001
0001
00001

Output
1
3
-1
0

2
1 2
```

Listing 16: one-shot example input for code revision

```
### Example 1
### TASK:
Polycarp has $n$ different binary words. A word called binary if it contains only characters '0' and '1'. For
    example, these words are binary: "0001", "11", "0" and "0011100".

Polycarp wants to offer his set of $n$ binary words to play a game "words". In this game, players name words
    and each next word (starting from the second) must start with the last character of the previous word.
    The first word can be any. For example, these sequence of words can be named during the game: "0101",
    "1", "10", "00", "00001".

Word reversal is the operation of reversing the order of the characters. For example, the word "0111" after
    the reversal becomes "1110", the word "11010" after the reversal becomes "01011".

Probably, Polycarp has such a set of words that there is no way to put them in the order correspondent to the
    game rules. In this situation, he wants to reverse some words from his set so that:  the final set of
    $n$ words still contains different words (i.e. all words are unique);  there is a way to put all words
    of the final set of words in the order so that the final sequence of $n$ words is consistent with the
    game rules.

Polycarp wants to reverse minimal number of words. Please, help him.
```

```
-----Input-----

The first line of the input contains one integer $t$ ($1 \le t \le 10^4$) -- the number of test cases in the
    input. Then $t$ test cases follow.

The first line of a test case contains one integer $n$ ($1 \le n \le 2\cdot10^5$) -- the number of words in
    the Polycarp's set. Next $n$ lines contain these words. All of $n$ words aren't empty and contains only
    characters '0' and '1'. The sum of word lengths doesn't exceed $4\cdot10^6$. All words are different.

Guaranteed, that the sum of $n$ for all test cases in the input doesn't exceed $2\cdot10^5$. Also, guaranteed
    that the sum of word lengths for all test cases in the input doesn't exceed $4\cdot10^6$.

-----Output-----

Print answer for all of $t$ test cases in the order they appear.

If there is no answer for the test case, print -1. Otherwise, the first line of the output should contain $k$
    ($0 \le k \le n$) -- the minimal number of words in the set which should be reversed. The second line
    of the output should contain $k$ distinct integers -- the indexes of the words in the set which should
    be reversed. Words are numerated from $1$ to $n$ in the order they appear. If $k=0$ you can skip this
    line (or you can print an empty line). If there are many answers you can print any of them.

-----Example-----
Input
4
4
0001
1000
0011
0111
3
010
101
0
2
00000
00001
4
01
001
0001
00001

Output
1
3
-1
0

2
1 2

### RELEVANT FUNCTIONS:

```module
def reverse_words(string):
    """
    description: this function reverses each word in the given string.
    input:
    string (str): the input string.
    output:
    reversed_string (str): the reversed string with each word reversed.
    """
    return ' '.join(s[::-1] for s in string.split(' '))
```

```module
def count_start_end_chars(words):
    """
    Description: This function counts the number of words that start and end with each character.
    Input:
    words (list): A list of binary words.
    Output:
    start_count (defaultdict): A dictionary containing the count of words that start with each character.
    end_count (defaultdict): A dictionary containing the count of words that end with each character.
    """
    start_count = collections.defaultdict(int)
    end_count = collections.defaultdict(int)
    for word in words:
        start_count[word[0]] += 1
        end_count[word[-1]] += 1
    return start_count, end_count
```
```

Listing 17: one-shot example output for normal code generation

```
### RESPONSE:

```python
```

```
for _ in range(int(input())):
    n = int(input())
    mass = []
    zo = 0
    oz = 0
    zz = 0
    oo = 0
    ozs = []
    zos = []
    ozss = set()
    zoss = set()
    for j in range(n):
        k = input()
        mass.append(k)
        if k[0] == '0' and k[-1] == '1':
            zoss.add(k)
            zos.append(j + 1)
            zo += 1
        elif k[0] == '1' and k[-1] == '0':
            ozss.add(k)
            ozs.append(j + 1)
            oz += 1
        elif k[0] == '0' and k[-1] == '0':
            zz += 1
        else:
            oo += 1
    if zz and oo and not oz and not zo:
        print(-1)
        continue
    else:
        if zo > oz:
            print((zo - oz) // 2)
            ans = []
            need = (zo - oz) // 2
            i = 0
            while need:
                zzz = mass[zos[i] - 1][len(mass[zos[i] - 1]) - 1:: -1]
                if zzz not in ozss:
                    ans.append(zos[i])
                    need -= 1
                i += 1
            print(*ans)
        else:
            print((oz - zo) // 2)
            ans = []
            need = (oz - zo) // 2
            i = 0
            while need:
                zzz = mass[ozs[i] - 1][len(mass[ozs[i] - 1]) - 1:: -1]
                if zzz not in zoss:
                    ans.append(ozs[i])
                    need -= 1
                i += 1
            print(*ans)
```

Listing 18: one-shot example output for modularized code generation/revision

```
STEP 1: GENERATE SUB-MODULES:
```module
def count_start_end_chars(words):
    """
    Description: This function counts the number of words that start and end with each character.
    Input:
    words (list): A list of binary words.
    Output:
    start_count (defaultdict): A dictionary containing the count of words that start with each character.
    end_count (defaultdict): A dictionary containing the count of words that end with each character.
    """
```

```module
def reverse_words(words, reversed_indices):
    """
    Description: This function reverses the specified words in the given list.
    Input:
    words (list): A list of binary words.
    reversed_indices (list): A list of indices indicating the words to be reversed.
    Output:
    reversed_words (list): A new list of words with the specified words reversed.
    """
```

STEP 2: GENERATE PYTHON CODE
```python
import collections

def count_start_end_chars(words):
    start_count = collections.defaultdict(int)
        end_count = collections.defaultdict(int)
        for word in words:
```

```
                start_count[word[0]] += 1
                end_count[word[-1]] += 1
        return start_count, end_count

def reverse_words(words, reversed_indices):
    reversed_words = []
    for i, word in enumerate(words):
        if i in reversed_indices:
            reversed_words.append(word[::-1])
        else:
            reversed_words.append(word)
    return reversed_words

def solve_task(words):
    start_count, end_count = count_start_end_chars(words)

    characters_with_difference = []
    for char in start_count:
        if abs(start_count[char] - end_count[char]) > 1:
            characters_with_difference.append(char)

    reversed_indices = []
    for char in characters_with_difference:
        difference = abs(start_count[char] - end_count[char])
        reverse_count = difference // 2
        if start_count[char] < end_count[char]:
            indices = [i for i, word in enumerate(words) if word.startswith(char)]
            reversed_indices.extend(indices[:reverse_count])
        else:
            indices = [i for i, word in enumerate(words) if word.endswith(char)]
            reversed_indices.extend(indices[:reverse_count])

    reversed_words = reverse_words(words, reversed_indices)

    total_reversed = len(reversed_indices)
    return total_reversed, reversed_words

t = int(input())
for _ in range(t):
    n = int(input())
    words = []
    for _ in range(n):
        words.append(input())
    total_reversed, reversed_words = solve_task(words)
    print(total_reversed)
    if total_reversed != 0:
        print(*reversed_words)

```
```

