# OpenReview forum: "CodeChain: Towards Modular Code Generation Through Chain of Self-revisions with Representative Sub-modules"
_ICLR.cc/2024/Conference — ICLR 2024 poster_

### Official Review · Reviewer_dD56 · 2023-10-30

**Soundness:** 3 good
**Presentation:** 3 good
**Contribution:** 2 fair
**Rating:** 6
**Confidence:** 3

**Summary:**

This paper proposes a new inference framework for modularised code generation using a large language model. Existing LLM tends to generate monolithic code blocks while a complex coding task usually needs to be broken down into multiple sub-tasks with each addressed by a sub-module. Motivated by the modular programming discipline, this paper proposes to iteratively ask LLM to generate code solutions with sub-modules and add the potentially valid sub-modules to the prompts for LLM’s generation at the next round. The sub-module selection is conducted by test case filtering and clustering.  The proposed framework yielded nontrivial improvements over the direct generation baselines.

**Strengths:**

In general, this paper follows the self-revision practice to improve the quality of code generated by LLM in a trial-and-error manner. However, instead of solely focusing on the functional correctness of the code, this paper raises the concern about LLM’s ability to decompose a complex coding task into multiple sub-tasks so as to generate reliable and reusable sub-modules. Although both task decomposition and chain-of-thoughts have been widely studied in different NLP applications, they are delicately adapted to fit the modular programming discipline in order to benefit code generation. Empowering LLM to produce modular code is not only helpful in improving the code’s reliability but also important for reducing the cost of subsequent manual maintenance. So this work’s originality and significance are considerable to me.

**Weaknesses:**

+ Sub-module filtering by public tests: given that the objects to be filtered are sub-modules generated by LLMs but we actually have no idea how the LLMs will decompose the target task/module into what sub-tasks/sub-modules. Then how can we compose the tests for the unknown sub-modules?

+ Four schemes for deciding the number of clusters are investigated. However, different tasks are of different complexity and it is difficult to decompose them into a fixed number of sub-tasks. Then is it possible that LLMs may generate some sub-modules which are never used in the task-level solution? If yes, then what are the effects of those sub-modules?

+ Kmeans are selected for clustering by why? Will density-based clustering like DBSCAN work better given it doesn’t need to specify the number of clusters as a prior?

+ As shown in Figure 5 and Figure 7, I failed to observe a consistent pattern to help me decide the optimal round of self-revision for CodeChain. The authors explained that the performance degradation at the 5th round is because of overfitting to the public test sets but this claim lacks support.

+ The title of the y-axis is missing in charts like Figures, 6, 7

**Questions:**

In table 2, why is the result of GPT4 (All: 34.75) is even bettrer than Self-repair+GPT4 with GPT4 as the feedback source (33.30)?

---

> ### Author Response · Authors · 2023-11-19
>
> Thank you for the positive review! We are glad that you appreciate our approach to adopt modular programming discipline and benefit code generation tasks. Please see our responses below:
>
> **Q1: Sub-module filtering by public tests - how can we compose the tests for the unknown sub-modules?**
>
> We apologize for the lack of clarity on this. Our filtering step selects all generated samples that have passed all public (seen) unit test cases. From the filtered samples, we collect all sub-modules to further do clustering. In most (approximately 70% with GPT3.5/GPT4 and about 50% on WizardCoder 13B) of the instances, at least one of the 20 generations passes all the public test cases so this conservative filtering does not significantly affect CodeChain. However, this is indeed an interesting suggestion, which opens up future research directions on generating sub-module descriptions along with unit-test cases as part of the planning step - which can lead to more effective filtering even from partially correct solutions.
>
> **Q2: Fixed number of clusters may entail having some generated sub-modules in the prompt, which will never be used in the task-level solution.  If yes, then what are the effects of those sub-modules?**
>
> We do not require the model to reuse these sub-modules in the prompt verbatim. In most scenarios, we observe that the model actually adapts or modifies these sub-modules to improve them and integrate them into the final solutions, rather than just copy the sub-modules word by word. So, to answer this question,  instead, we checked whether the function header is re-used verbatim from the prompt to the model’s generation.
> Over the subset of APPS-test used in Table 2 (with 20 generations per task), we saw only in ~13% of generations there is at least one sub-module that is not re-used verbatim. And in those cases, on average the model also generates around 1.43 new sub-modules (that are not present in the prompt).
> Inspecting into these we saw the model is indeed paraphrasing the function names (like `process_queries` -> `ProcessQueries`
> and `compute_square_vertices` -> `calculate_square_vertices`) but their functionality is still very similar.
> Among such 13% of generations, both the average modularity and reusability (by GPT4) ratings are around 3 (which is still much higher than that of direct generation).
>
> **Q3: DBScan (which dynamically determines cluster size)  VS KMeans clustering**
>
> In Figure 6 we compared different strategies of clustering where dynamic refers to the dynamic number of clusters selected based on the silhouette coefficients. Based on this analysis we concluded that the best strategy is to use a descending order of clustering, as particularly in the early iterations, it seems to be most effective. But the descending strategy also resonates with the well-established exploration to exploitation paradigm, as the models become more confident in their generation. For DBScan, we still need an understanding of the densities of modular functions to be able to cluster them efficiently. As the coding problems are typically very different, clusters of generated modular functions will likely not have similar densities for DBScan to work well.
>
> **Q4: I failed to observe a consistent pattern to help me decide the optimal round of self-revision for CodeChain.**
>
> Apologies for the lack of clarity. In Figure 5, the optimal rounds of self-revision might differ due to the different complexity of problems. However, as we see in Figure 6, the overall performance across all samples achieves its peak at round 4. In Figure 7, this trend is clearer in larger-size models (7B, 15B, and 34B) but smaller models do not exhibit this trend due to their weak foundational representations. Moreover, on page 16 (Appendix Section D “Using public tests as a proxy to select optimal revision step.“) we prescribe using the public tests as a proxy to gauge model performance throughout the chain of self-revisions and deciding the optimal revision step. Therefore, in practice, we can dynamically decide the optimal round of self-revision given a specific problem and some public test cases rather than relying on validation set results
>
> **Q5: In Table 2, why is the result of GPT4 (All: 34.75) is even better than Self-repair+GPT4 with GPT4 as the feedback source (33.30)?**
>
> We would like to note that the difference is not that significant and well within the expected variance that is naturally observed in a sampling-style generation. Another potential reason in this case can be due to (1) different versions of GPT4 used in the Self-repair and our work and (2) different instruction prompts between Self-repair and our work.  On the other hand, the performance boost achieved by CodeChain is indeed very significant (34.75 -> 61.5).
>
> **Q6: the title of the y-axis is missing in charts like Figure 6, 7**
>
> Thanks for pointing this out! We will fix the Figures and thoroughly check the paper in our final revision.

---

> ### Author Response · Authors · 2023-11-22
> **Reminder for discussion/ additional questions**
>
> Dear Reviewer,
>
> Thank you again for all the feedback on our paper! Please do let us know if you still have any questions before the discussion period ends.
>
> Regards,

---

> > ### Comment · Reviewer_dD56 · 2023-11-23
> >
> > thanks for the clarification and the additional materials provided, most of my concerns are properly addressed and I'm happy to keep my initial rating.

---

### Official Review · Reviewer_hL3y · 2023-11-01

**Soundness:** 3 good
**Presentation:** 3 good
**Contribution:** 2 fair
**Rating:** 6
**Confidence:** 3

**Summary:**

This paper proposes a modular code generation approach for complex programming tasks. CodeChain extracts and clusters generated sub-modules, selects representative implementations and instructs the LLM to generate new solutions using these selected modules. Experimental results show that CodeChain significantly improves modularity and correctness, achieving relative pass@1 improvements of 35% on APPS and 76% on CodeContests. The framework works well with both OpenAI LLMs and open-sourced LLMs like WizardCoder. The paper also includes comprehensive ablation studies that provide insights into CodeChain's performance.

**Strengths:**

* SoTA performance on code generation benchmarks
* CodeChain works well across LLMs (GPT-3.5, GPT-4 and WizardCoder)
* The paper is well-written and is easy to understand.
* The authors provided extensive ablation.

**Weaknesses:**

* Correctness/Soundness:
(1) Programs generated by CodeChain are with high levels of modality and reusability on Likert scale judging by GPT-4 prompt. It is unclear how this evaluation align with human preference.
(2) The effectiveness of sub-module generation is unclear.
(3) The analysis on the chain of self-revisions sees a slight performance drop in the 5th iteration, which hints at the limitation of self-revise prompting.

* Novelty/Originality:
1. Using CoT to generate demonstrations and choosing representative demonstrations has been explored in [1]. The novelty of this work lies in employing this idea for code generation, which is incremental.
2. Missing citation: [2] explores a similar idea of decomposing source code into components.
3. The idea of utilizing LLMs' ability to self-revise has been studied in [3,4] and more.

* Writing could be improved: There are multiple references to an Appendix section (e.g., Appendix F) without clarifying which figure/prompt is being referred to.

References
[1] Automatic Chain of Thought Prompting in Large Language Models, ICLR 2023
[2] Outline, Then Details: Syntactically Guided Coarse-To-Fine Code Generation, ICLM 2023
[3] Large Language Models Can Self-Improve, ICLR 2023
[4] CRITIC: Large Language Models Can Self-Correct with Tool-Interactive Critiquing

**Questions:**

In section 3.2, the author mentioned, "We append the instruction with a one-shot demonstration.", which part of the referenced figure (Figure 3) or the appendix (Appendix F) is the one-shot demonstration? How is this demonstration being constructed?



=== Post-rebuttal ====
Thanks for the detailed clarification and the additional evaluation. While I still have concerns regarding the novelty and the proper evaluation with GPT-4 (in terms of human evaluation), I think this paper is a good contribution to the community. I increased my score.

---

> ### Author Response · Authors · 2023-11-19
>
> Thank you for your reviews! We are glad that you appreciated our extensive experiment results with the SoTA performance of CodeChain and its benefits across LLMs. Please find below our responses:
>
> **Q1: How does GPT-4 evaluation align with human preference**
>
> Thank you for this suggestion! We are currently conducting a human evaluation survey to estimate the human preference between CodeChain and related baselines. We will release the human evaluation results as soon as possible.
>
> **Q2: The effectiveness of sub-module generation is unclear.**
>
> We would like to highlight that sub-module generation can indeed affect the performance as compared to direct generation. One reason was that most of the existing LLMs were not pretrained/ fine-tuned with perfectly modularized code data. However, to improve code reusability and refinement, it is essential to promote more modularity in code generation. We explained this point in Section 3.2 and Table 3. From the ablation results in Table 3, we can see that despite the initial performance drop with the sub-module generation, refining these modules iteratively eventually led to the best performance as compared to monolithic code generation.
>
> **Q3: Performance drop in 5th iteration for APPS  which hints at the limitation of self-revise prompting.**
> As explained on page 8 ( section “Analysis on chain of self-revisions”), the drop in the 5th iteration for APPS is likely due to the selected samples for refining becoming overfit to the small number of public unit tests. We also observe that for the CodeContests dataset, the public test cases are much more diverse and extensive, and indeed in that case the performance of CodeChain continues to improve till the 5th iteration. We also prescribe using the public tests as a proxy to gauge model performance throughout the chain of self-revisions and decide the optimal revision step. (Appendix Section D, page 16, “Using public tests as a proxy to select optimal revision step”)
>
>
> **Q4: Novelty is incremental**
>
> We thank the reviewer for the missing references. We will incorporate these references in our final revision. However, we would like to point out the specific aspects of novelty, both in our design of CoT and the refinement strategy, which sets us apart from these counterparts:
> - *Comparison to Auto-CoT [1] and ChainCoder [2]:* CodeChain is distinct from these approaches in 2 major ways:
>
>   - CodeChain emphasizes generating modular solutions, a key feature in coding that’s rarely explored. By decomposing the task into logical sub-modules, it achieves some sort of high-level planning that is essential for complex problem-solving. ChainCoder, on the other hand, focuses on sub-sequence planning and still generates only a single monolithic block of code.
>
>   - Although Auto-CoT uses clustering to construct representative CoT few-shot demonstrations, CodeChain uses clustering for a very different purpose: select representative sub-modules to reuse/refine themselves iteratively. This objective is very different from Auto-CoT and it is not trivial to adapt Auto-CoT in this context. Moreover, CodeChain only relies on its own generation samples, without needing to have a data source to retrieve CoT demonstrations like Auto-CoT.
>
> - *Comparison to other self-refinement techniques such as [3, 4]:* Other self-refinement strategies are limited by the fact that they can only use feedback from individual generations to independently refine each of them. Whereas, CodeChain is a first-of-a-kind self-refinement strategy that learns to refine the generated code by looking at collective insights drawn from all of its past generations and their components (sub-modules). Our significant performance improvements over other self-refinement techniques categorically establish that accumulating knowledge from multiple generations and their modular components can indeed lead to better and more accurate refinement.
>
> References:
>
> *[1] Automatic Chain of Thought Prompting in Large Language Models, ICLR 2023*
>
> *[2] Outline, Then Details: Syntactically Guided Coarse-To-Fine Code Generation, ICLM 2023*
>
> *[3] Large Language Models Can Self-Improve, ICLR 2023*
>
> *[4] CRITIC: Large Language Models Can Self-Correct with Tool-Interactive Critiquing*
>
> **Q5: In section 3.2, the author mentioned, "We append the instruction with a one-shot demonstration.", which part of the referenced figure (Figure 3) or the appendix (Appendix F) is the one-shot demonstration? How is this demonstration being constructed?**
>
> Apologies for the lack of clarity. On Page 6, we mentioned that “ we randomly selected this one-shot sample from the APPS training split (see Appendix G).” Please refer to Appendix G for a copy of the one-shot example we used. We will thoroughly check the manuscript in the final revision to improve the paper's clarity.

---

> ### Author Response · Authors · 2023-11-21
> **How does GPT-4 evaluation align with human preference**
>
> Thank you for waiting! To answer this question, we conducted the human evaluation following your suggestion and updated the paper. Please refer to Appendix D "Human evaluation results" for our results and analysis in detail. In summary, we found that compared to the GPT-4 based evaluation results, the human evaluation results of CodeChain are slightly different with more even distributions across the upper range (score 3-5 in both modularity and reusability qualities). However, we still observed a consistent improvement when comparing CodeChain against the traditional direct generation approach. This observation is well aligned with our previous findings from the GPT-4-based evaluation results.

---

> > ### Comment · Reviewer_hL3y · 2023-11-22
> > **Re: GPT-4 evaluation align with human preference**
> >
> > Thanks for your effort in providing the evaluation.
> >
> > From Figure 9 (main paper) and Figure 16 (appendix), is the alignment with human preference not as good as that of GPT4?
> > In particular, from my understanding, Figure 16 shows that human preference has more medium scores than very-high scores compared to those evaluated by GPT4.

---

> > > ### Author Response · Authors · 2023-11-22
> > >
> > > Thank you for your response! This is true, the results indicate that human annotators are more conservative and have stricter standards, hence the lower average scores than GPT4 results. This is consistent with some recent works [1, 2] which observed the potential biases in GPT-based evaluation. Yet, these papers still found a reasonable level of alignment between GPT-based evaluation and human evaluation and hence, still advocate for these automatic evaluation methods.
> > >
> > > Looking more closely at our results (figure 16), to judge the modularity aspect of CodeChain samples, we found that R3 & R5 clearly > R0. But for re-usability, CodeChain R3 > R0, while R5 is in between R0 and R3. This may indicate a sign of overfitting that explains some performance drop in R5 in terms of passing rate. Overall, either by GPT-4 evaluation (figure 9) or human evaluation (figure 16), our finding about the improvement of CodeChain by modularity/ reusability qualities against the baseline still stands.
> > >
> > > We will update more analysis related to the discussion here in our final revision.
> > >
> > > References:
> > >
> > > *[1] Judging LLM-as-a-Judge with MT-Bench and Chatbot Arena*
> > >
> > > *[2] Can Large Language Models Be an Alternative to Human Evaluations?*

---

> ### Author Response · Authors · 2023-11-23
>
> Dear Reviewer,
>
> Thank you for all the feedback and suggestions to improve our paper! From our responses and follow-up discussions, we hope the reviewer can consider raising the final rating score of the paper. Please feel free to let us know any concerns or questions you may have.
>
> Regards,

---

### Official Review · Reviewer_sbrc · 2023-11-03

**Soundness:** 3 good
**Presentation:** 3 good
**Contribution:** 2 fair
**Rating:** 8
**Confidence:** 4

**Summary:**

The paper proposes CodeChain a method for prompting LLMs to generate modular code and reuse the generated submodules in subsequent iterations of prompting. The generated submodules are extracted and clustered to find representative and reusable components for iterations of self-revision. Experimental results on the APPS and CodeContests demonstrates that CodeChain significantly improves pass@1 metric when compared to several prior approaches.

**Strengths:**

* The CodeChain prompting technique proposed in the paper is intuitive and easy to utilize. Designing a prompting strategy for inducing and leverage modular functions seems novel.
* The experimental results on the selected benchmarks show the effectiveness of the prompting strategy relative to several prior methods.
* Studies on the impact of different clustering strategies, embedding choices and revision sampling are informative.
* Overall CodeChain seems like a good prompting strategy which is simple and effective.

**Weaknesses:**

* The clustering strategy seems to add only a small performance improvement to the overall approach. Randomly picking the modules also seems to do reasonably well (Table 3). One experiment that would be helpful is adding all submodules instead of randomly picking the generated modules to include in future revisions. Would this eliminate the need for clustering.

* It is unclear how much the prompting strategy is sensitive to specific wording of the prompt and alternative formulations. Did the authors try multiple variants of the prompt and if so what was the variance and sensitivity of the results. Is it possible that there are prompts which could instruct the model to do revisions in a single shot? Could this improve direct generation?

 * The evaluation largely relies on two benchmarks. It would be good to see the evaluation extend to some of the appropriate subsets in https://github.com/bigcode-project/bigcode-evaluation-harness

**Questions:**

* The authors observe that the training datasets do not filter for modularity. Have the authors tried filtering the training dataset for modular code and do light-weight fine-tuning with parameter efficient tuning methods?

---

> ### Author Response · Authors · 2023-11-19
>
> Thank you for the positive reviews! We are glad you found our approach intuitive and effective with informative experiment results. Please find our responses below:
>
> **Q1: Clustering Strategy adds only small performance improvement. Randomly picking the modules also seems to do reasonably well.**
>
> We beg to differ on one aspect of this comment - While randomly choosing submodules seems to do reasonably well on the Introductory and Interview categories, in the complex category (Competition) it is at par with the naive (direct) code generation. But strategically choosing the representatives works well across all categories, improving the average by around 7% over direct code generation.
>
> **Q2: One experiment that would be helpful is adding all submodules instead of randomly picking the generated modules to include in future revisions. Would this eliminate the need for clustering.**
>
> This approach would require adding all generated sub-modules from all previously generated samples. This will significantly increase the size of the input context and hence, increase the compute cost/ financial cost. Further, it can also distract the model with a long context of sub-modules, including even the poor quality ones or ones that are not very reusable. Compared to this approach, our simple clustering-sampling approach serves as an effective way of selecting the most representative and reusable sub-modules and hence boosting the performance of all LLMs.
>
> **Q3: Did the authors try multiple variants of the prompt and if so what was the variance and sensitivity of the results.**
>
> Indeed we experimented with different versions of prompts, including prompts that instruct the model to perform traditional chain-of-thought reasoning (following prompt templates from https://arxiv.org/abs/2205.11916, https://arxiv.org/abs/2201.11903, and related work). However, we found that most of the current models performed poorly with these conventional CoT prompting on benchmarks such as APPS. For instance, using GPT3.5 and a variant of the traditional CoT prompt, we observed the following pass@1 on the APPS validation set: Introductory=25%, Interview=19.7%, Competition=6.6%. These results are 3-8 points lower than our CodeChain prompt. One reason for this observation is that these CoT prompts are more appropriate for reasoning tasks in which the thought process could be easily explained in natural language. For coding problems in general, it is still very challenging (even for humans) to explain complex coding solutions in a natural step-by-step thought process. Therefore, our final prompt is a two-step generation (sub-modular abstraction then final implementation) to adapt CoT to the coding domain while not compromising the model performance too much.
>
> **Q4: Is it possible that there are prompts which could instruct the model to do revisions in a single shot? Could this improve direct generation?**
>
> Even assuming some prompting strategy is indeed able to effectively instruct self-revision in a single-shot manner, this and other self-refinement strategies are limited by the fact that they can only use feedback from individual generations to independently refine each of them. Whereas, CodeChain is a first-of-a-kind self-refinement strategy that learns to refine the generated code by looking at collective insights drawn from all of its past generations. Our significant performance improvements over other self-refinement techniques categorically establish that accumulating knowledge from multiple generations can indeed lead to better and more accurate refinement.

---

> ### Author Response · Authors · 2023-11-19
>
> **Q5: The evaluation largely relies on two benchmarks APPS and CodeContest. Why not evaluate on appropriate subsets of bigcode-evaluation**
>
> Most of the code-generation tasks in the bigcode-evaluation are on HumanEval (and its variants), MBPP, APPS, DS-1000, Math reasoning (GSM-8K Python), and CoNaLa. HumanEval, MBPP, DS-1000, and Math Reasoning typically involve simpler tasks (more atomic/simpler in nature than the simplest categories of APPS) where modular code generation is not really required. CoNaLa on the other hand does not have unit-test cases to evaluate correctness and relies on BLEU-based evaluation which has some well-known limitations and bias.
>
> We would also like to emphasize that Generating modular code is a very hard task and this is the first work to address modular code generation.
> - *Needs to be showcased on challenging/complex tasks:* our choice of datasets has been based on our need to showcase CodeChain on the hardest categories of competitive programming problems where modular generation becomes even more essential. The datasets we showcase on APPS and CodeContest (and also LeetCodeHardGym, results in Appendix Section D, section “Results on LeetCodeHardGym “) are some of the hardest benchmarks for code generation and research on this has been relatively limited. Whereas datasets like HumanEval and MBPP have already been relatively saturated, with SoTA models achieving over 90% already.
> - *A lack of pretraining data having modular code:* most of the pretraining data on which these models are trained are from GitHub and other sources, where the majority of the code would not be following a perfectly modularized style, thus making this a hard task to learn.
>
> **Q6: The authors observe that the training datasets do not filter for modularity. Have the authors tried filtering the training dataset for modular code and do light-weight fine-tuning with parameter efficient tuning methods?**
>
> We thank the reviewer for the suggestion. We believe this kind of manual filtering would not be possible: based on our study on datasets like APPS, our understanding is that very few of the training data would have such modular characteristics, at least in the ground truth solutions. Given this, we believe having a lightweight inferencing (training-free) paradigm (only using 20 generations and a few rounds of iterations) which still brings a significant boost in performance and modularity, is indeed a more practical and efficient way for modular code generation.
>
> However, having generated modular code in CodeChain style using GPT-4 or WizardCoder (WizardCoder34B) models, we can indeed generate synthetic finetuning data and distill this knowledge into open-source models (like smaller WizardCoder models). This, we believe, is an important future direction to follow.

---

> ### Author Response · Authors · 2023-11-22
> **Reminder for discussion/ additional questions**
>
> Dear Reviewer,
>
> Thank you again for all the feedback on our paper! Please do let us know if you still have any questions before the discussion period ends.
>
> Regards,

---

> > ### Comment · Reviewer_sbrc · 2023-11-22
> > **Reply to authors**
> >
> > Thank you for the detailed responses to my questions. I think the paper would be interesting to the community and therefore raising my score.

---

### Official Review · Reviewer_3rJJ · 2023-11-09

**Soundness:** 3 good
**Presentation:** 3 good
**Contribution:** 3 good
**Rating:** 6
**Confidence:** 2

**Summary:**

This paper proposes the CodeChain approach for code generation in LLM (Large Language Model) sub-modules, aiming to enhance the modularity and accuracy of the resulting code.

**Strengths:**

The paper is well-structured and presents its arguments in a clear fashion.

The prompt method described is uncomplicated yet efficacious.

The experimental evaluation encompasses both closed-source and open-source models, providing a comprehensive analysis.

**Weaknesses:**

It is unclear whether the method can improve the quality of code generation specifically for Codellama models.

The potential of CodeChain to bolster problem-solving capabilities in other domains, such as mathematics, is not established.

**Questions:**

See Weaknesses

---

> ### Author Response · Authors · 2023-11-19
>
> Thank you for the positive review! We are glad you found our proposed approach efficacious with comprehensive experiment and analysis. Please see our responses to your questions below:
>
> **Q1: Is CodeChain effective for Codellama:**
>
> Both WizardCoder-7B and WizardCoder-34B are the instruction-tuned models from Codellama-7B and Codellama-34B. Our results of these models in Figure 7 show a huge performance improvement (for example, from 9% to 18% on 34B model) in pass@1 results on the APPS-validation set. To apply the original instruction tuned Codellama, we expect a similar trend of performance improvement (after adapting our prompt to suit the Codellama instruction style).
>
> **Q2: CodeChain on Math Reasoning:**
>
> Thank you for this suggestion! Since our focus is on modular code generation, we found that most existing math reasoning (like GSM8K-Python) problems are not a good fit for this objective, since solving these problems will not typically require more than one non-trivial module. Even in competitive programming, CodeChain achieves the biggest boost in the most complex categories, which really demand decomposition into multiple submodules.

---

> ### Author Response · Authors · 2023-11-22
> **Reminder for discussion/ additional questions**
>
> Dear Reviewer,
>
> Thank you again for all the feedback on our paper! Please do let us know if you still have any questions before the discussion period ends.
>
> Regards,

---

### Public Comment · ~Dong_HUANG4 · 2023-11-11
**Inquiring for the comparsion with baselines.**

Hi, thank you for submitting your paper to ICLR. I find the paper commendable, particularly in its approach of enhancing code generation through the use of the chain of thought—a sound strategy.

However, I'm curious about why there isn't a comparison with prior strategies that also leverage the chain of thought for code generation, such as CodeCoT. Additionally, I noticed there's no comparison with state-of-the-art self-refine code generation strategies like Self-Edit, Self-refine, and Self-debug.

I've observed that you discuss the differences between your strategies and recent self-refine approaches, but I'm interested in seeing actual comparison results. Could you provide insights into this aspect?

---

> ### Author Response · Authors · 2023-11-19
>
> Thank you for your comments! We are glad you found our paper a commendable approach for improving code generation. Please see our responses below:
>
> **Comparison to CodeCOT, Self-Edit, Self-refine, and Self-debug:**
>
> In Figure 5, we have compared our approach with *Self-debug* and *Reflexion* (which is one of the latest papers on iterative self-refinement). In Table 2, we also compared with *Self-repair*. In these experiments, we observe that CodeChain can outperform all of these baselines consistently.
>
> *CodeCOT.* Although CodeChain and CodeCOT both use chain-of-thought techniques (a general prompting technique indeed), the focus of CodeChain is quite distinct from CodeCOT: we aim to generate modularized codes by decomposing a monolithic code solution to representative sub-modules. In CodeCOT, the CoT technique is only for the model to explain its reasoning process in natural language without modularizing the code by high-level logical steps. This approach showed good performance on simple tasks in HumanEval but might not work well in more complex coding tasks in APPS. In our preliminary results, using a similar version of prompting to CodeCOT actually led to a significant performance drop in APPS.
>
> *Self-Edit* requires finetuning a new model as the Code Editor. To make fair comparisons to CodeChain, we mainly replicate related methods that do not require additional finetuning. For completeness, we included the results of Self-edit in Table 1.
>
> *Self-refine* focuses on specific tasks like Sentimental Reversal, Dialogue Response, Code Optimization, and Code Readability Improvement. Adapting Self-refine to a very different task like text-to-code generation on APPS is not trivial and would require making significant modifications to the original method. For instance, in the Code Readability Improvement task, Self-refine mostly focuses on specific aspects of the code like variable names and comments. These are very different and not enough for code generation tasks e.g. APPS, which considers functional correctness as the main metric.

---

### Author Response · Authors · 2023-11-21
**Reminder for discussion/ additional questions**

Dear reviewers,

Thank you again for all the informative and constructive feedback! We truly appreciate all the suggestions from the reviewers to improve this work. We have revised the paper and responded to all questions from the reviewers. As the discussion period is ending soon, please do let us know whenever you have any more concerns or would like to discuss further about the paper.

Regards,

---

### Meta-Review · Area_Chair_KJUy · 2023-12-14

**Metareview:**

> The paper proposes CodeChain a method for prompting LLMs to generate modular code and reuse the generated submodules in subsequent iterations of prompting. The generated submodules are extracted and clustered to find representative and reusable components for iterations of self-revision. Experimental results on the APPS and CodeContests demonstrates that CodeChain significantly improves pass@1 metric when compared to several prior approaches.

The experimental scope is somewhat limited, and the paper is not the first to propose this kind of approach. But overall, this is a sound study of inference-time self-improvement for code generation. The authors were quite active in the discussion period, and all reviewers agree: this is a worthwhile addition to ICLR's lineup.

**Justification For Why Not Higher Score:**

The experimental scope is somewhat limited, and the paper is not the first to propose this kind of approach.

**Justification For Why Not Lower Score:**

N/A

---

### Decision · Program_Chairs · 2024-01-16

Accept (poster)